# QP-SNNs: Quantized and Pruned Spiking Neural Networks

**Wenjie Wei**[1], **Malu Zhang**[1],[*] **Zijian Zhou**[1], **Ammar Belatreche**[2],
**Yimeng Shan**[3], **Yu Liang**[1], **Honglin Cao**[1], **Jieyuan Zhang**[1], **Yang Yang**[1]

[1]University of Electronic Science and Technology of China
[2]Northumbria University, [3]Liaoning Technical University

## Abstract

Brain-inspired Spiking Neural Networks (SNNs) leverage sparse spikes to encode information and operate in an asynchronous event-driven manner, offering a highly energy-efficient paradigm for machine intelligence. However, the current SNN community focuses primarily on performance improvement by developing large-scale models, which limits the applicability of SNNs in resource-limited edge devices. In this paper, we propose a hardware-friendly and lightweight SNN, aimed at effectively deploying high-performance SNN in resource-limited scenarios. Specifically, we first develop a baseline model that integrates uniform quantization and structured pruning, called QP-SNN baseline. While this baseline significantly reduces storage demands and computational costs, it suffers from performance decline. To address this, we conduct an in-depth analysis of the challenges in quantization and pruning that lead to performance degradation and propose solutions to enhance the baseline's performance. For weight quantization, we propose a weight rescaling strategy that utilizes bit width more effectively to enhance the model's representation capability. For structured pruning, we propose a novel pruning criterion using the singular value of spatiotemporal spike activities to enable more accurate removal of redundant kernels. Extensive experiments demonstrate that integrating two proposed methods into the baseline allows QP-SNN to achieve state-of-the-art performance and efficiency, underscoring its potential for enhancing SNN deployment in edge intelligence computing.

## 1 Introduction

Inspired by the information processing paradigm of biological systems, Spiking Neural Networks (SNNs) encode information via binary spikes and process them in a sparse spike-driven manner (Gerstner & Kistler (2002); Izhikevich (2003)). This paradigm simplifies the matrix computations of weight and spike activity in SNNs from computationally intensive multiply-accumulate (MAC) operations to computationally efficient accumulate (AC) operations. Therefore, SNNs are regarded as a promising energy-efficient solution for achieving next-generation machine intelligence (Pfeiffer & Pfeil (2018); Roy et al. (2019b)). Furthermore, the energy efficiency of SNNs has driven the development of neuromorphic hardware, such as SpiNNaker (Painkras et al. (2013)), TrueNorth (Akopyan et al. (2015)), Loihi (Davies

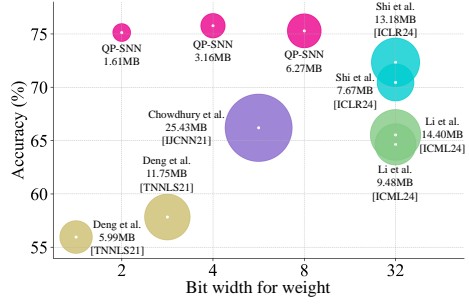

Figure 1: Comparison of QP-SNN and related work on CIFAR-100. The bubble size represents the model size.

et al. (2018)), Tianjic (Pei et al. (2019)) etc. These neuromorphic hardware can fully exploit the energy efficiency potential of SNNs. Despite these advantages, the application scenarios and performance of SNNs still require improvement compared to traditional artificial neural networks (ANNs).

---

[*]Corresponding author: maluzhang@uestc.edu.cn

In the past few years, the SNN community has focused primarily on designing complex SNNs architectures (Yu et al. (2020); Yao et al. (2024b;a); Luo et al. (2024); Wang et al. (2024b); Zhang et al. (2024)) and training strategies (Zhang et al. (2018; 2019; 2021a); Wu et al. (2021); Guo et al. (2022); Xu et al. (2023); Wei et al. (2024b)) to achieve impressive performance across various application tasks. While these studies have yielded satisfactory performance, they typically come at the cost of large model parameters, high memory consumption, and increased computational complexity (Yan et al. (2024); Liu et al. (2024); Cao et al. (2025)). This undermines the inherent energy efficiency of SNNs and restricts their applicability in resource-limited scenarios. To achieve effective deployment, a growing number of researchers have worked on compressing large-scale SNNs. Existing methods to improve SNN energy efficiency primarily fall into two categories: (1) reducing the precision of parameter representations and (2) reducing redundant parameters within the model.

Quantization is a key technique for the first category, which reduces memory storage and computational complexity by storing full-precision values in low bit-width precision (Gholami et al. (2022)). Based on whether an equal-size interval is used to discretize full-precision values, it can be divided into non-uniform and uniform quantization (Rokh et al. (2023)). Non-uniform quantization divides discretization intervals unevenly, enabling a more precise capture of critical information and leading to improved performance. However, it is challenging to deploy this approach on general computing hardware efficiently (Cai et al. (2017); Kulkarni et al. (2022)). In contrast, uniform quantization maps full-precision values to equal-sized discrete intervals, offering advantages such as simplicity, low computational cost, and efficient mapping to hardware (Zhu et al. (2016); Jain et al. (2020)).

Pruning is one of the effective methods for reducing redundant parameters in a model, which reduces the model size by removing unimportant connections (Li et al. (2016)). Pruning can be classified into unstructured and structured pruning (Vadera & Ameen (2022)). Unstructured pruning removes individual nodes like a single neuron of networks, resulting in unstructured sparsity. This often leads to a high compression rate, but requires specialized hardware or library support for acceleration (Han et al. (2015)). In contrast, structured pruning removes entire convolutional filters, ensuring model's structure. This avoids complex sparse matrix operations, enabling acceleration with standard hardware by taking advantage of a highly efficient library (He & Xiao (2023); Xu et al. (2020)).

Real-world deployments are typically limited by size, weight, area, and power. This makes combined quantization and pruning a promising approach for maximizing SNN compression. Existing research on integrating quantization and pruning in SNNs faces two challenges. Firstly, they do not sufficiently account for hardware-friendliness, for example, (Rathi et al. (2018)) and (Deng et al. (2021)) adopt unstructured pruning. Secondly, despite significant energy efficiency, these studies suffer from severe performance degradation, with evaluations limited to simple datasets (Chowdhury et al. (2021); Deng et al. (2021)). Thus, integrating both techniques for maximal compression while ensuring hardware efficiency and high performance remains challenging. In this paper, we introduce the QP-SNN, tailored for effective deployment in resource-limited environments. We first build a QP-SNN baseline that integrates uniform quantization and structured pruning. While this baseline offers notable efficiency gains, it suffers from reduced performance. To address this, we investigate the root causes of performance degradation in quantization and pruning, and propose solutions to enhance the baseline's performance. As shown in Figure 1, QP-SNN utilizing the proposed solutions achieves excellent accuracy and model size. We summarize main contributions as,

- We first develop a hardware-efficient and lightweight QP-SNN baseline by integrating uniform quantization and structured pruning. This baseline significantly reduces storage and computational demands, but suffers from performance limitations.

- To improve performance through quantization, we reveal that the vanilla uniform quantization in the QP-SNN baseline constrains the model's representation capability due to inefficient bit-width utilization. To address this, we propose a weight rescaling strategy (ReScaW) that optimizes bit-width usage for improved representation.

- To further boost performance through pruning, we introduce a novel structured pruning criterion for the QP-SNN baseline that leverages the singular value of spatiotemporal spike activity (SVS). This SVS criterion provides greater robustness across varying input samples and allows more precise removal of redundant convolutional kernels.

- Extensive experiments show that integrating ReScaW-based quantization and the SVS-based criterion into the baseline allows QP-SNN to achieve state-of-the-art performance and efficiency, revealing its potential for advancing edge intelligence computing.

## 2 RELATED WORK

**Quantization technique**   Early research on quantization in SNNs is primarily based on ANN-to-SNN conversion algorithms, where a quantized ANN is first trained and then converted into the corresponding quantized SNN version (Sorbaro et al. (2020); Roy et al. (2019a)). To mitigate the performance loss associated with the conversion, researchers have proposed many strategies, such as the utilization of activation penalty term (Sorbaro et al. (2020)) and the weight-threshold balancing method (Wang et al. (2020)). However,these quantized SNNs still experience significant performance degradation and long latency issues. To address these limitations, some studies have explored directly training quantized SNNs and introduced different strategies to enhance performance, such as alternating direction method of multipliers (Deng et al. (2021)), accuracy loss estimator (Pei et al. (2023)), suitable activation function (Hu et al. (2024)), and weight-spike dual regulation (Wei et al. (2024a); Wang et al. (2024a)). Despite performance improvement, these studies fail to effectively leverage the allocated bit-width, resulting in the limited expressive capability of models. Therefore, there still remains significant room for performance improvement.

**Pruning technique**   Existing research on pruning SNNs can be broadly divided into two groups. The first group is unstructured pruning. For example, (Yin et al. (2021)) use a magnitude-based method to remove insignificant weights, and (Shi et al. (2024)) propose a fine-grained pruning framework that integrates unstructured weight and neuron pruning to enhance SNN energy efficiency. Additionally, there are some biologically inspired unstructured pruning works (Bellec et al. (2017); Chen et al. (2022)). While these studies achieve great sparsity and performance, they lead to irregular memory access in forward propagation, requiring specialized hardware for acceleration. The second group is structured pruning that offers better hardware compatibility. (Chowdhury et al. (2021)) use principal component analysis on membrane potentials to evaluate channel correlations and eliminate redundant ones. However, it suffers from long latency and cannot handle neuromorphic datasets. Recently, (Li et al. (2024)) evaluate the importance of kernels based on spike activity, advancing the performance of pruned SNNs to a new level. However, this evaluation criterion exhibits high dependency on inputs and may not accurately reflect the importance of kernels.

**Compression with joint quantization and pruning**   Several studies have explored combining quantization and pruning to maximize the compression of SNNs. First, (Rathi et al. (2018)) adopt the STDP learning rule and a predefined pruning threshold to remove insignificant connections, and then quantizes retained important weights. Then, (Chowdhury et al. (2021)) perform principal component analysis on membrane potentials for spatial pruning and gradually decreases the time step during training for temporal pruning. They also use post-training quantization to compress retained weights. Moreover, (Deng et al. (2021)) formulate pruning and quantization as a constraint optimization problem in supervised learning, and address it with the alternating direction method of multipliers. However, these existing studies combining quantization and pruning face two main problems. Firstly, the unstructured pruning methods in (Rathi et al. (2018)) and (Deng et al. (2021)) require specialized hardware for efficient acceleration. Secondly, (Rathi et al. (2018)) only evaluate their method on very simple datasets, and (Chowdhury et al. (2021)) and (Deng et al. (2021)) only extend their methods to CIFAR (88.6% and 87.84% accuracy on CIFAR-10 with 5, 3 bits respectively), leading to significant room for improvement in both performance and efficiency.

## 3 QUANTIZED AND PRUNED SNN BASELINE

In this section, we develop the QP-SNN baseline by combining uniform quantization and structured pruning. These two compression techniques are highly compatible with existing hardware accelerators, significantly improving the model's deployment efficiency.

**Neuron model.**   Many spiking neurons have been proposed (Hodgkin & Huxley; Zhang et al. (2021a); Wei et al. (2023)), we use the Leaky Integrate-and-Fire (LIF) model in QP-SNN due to its high computational efficiency (Wu et al. (2018)). The membrane potential of LIF model is,

$$\tilde{\mathbf{U}}^l[t] = \tau\mathbf{U}^l[t-1] + \mathbf{X}^l[t], \tag{1}$$

where $\mathbf{U}^l[t]$ is the membrane potential of neurons in layer $l$ at time $t$, $\tau$ is the leaky factor, and $\mathbf{X}^l[t] = \mathbf{W}^l\mathbf{S}^{l-1}[t]$ is input current. Neurons integrate incoming signals and generate a spike when

the membrane potential exceeds the threshold $\theta$. The spike generation function is computed as,

$$\mathbf{S}^l[t] = \begin{cases} 1, & \text{if } \mathbf{U}^l[t] \geq \theta, \\ 0, & \text{otherwise.} \end{cases} \tag{2}$$

After spike emission, we use the hard reset mechanism to update the membrane potential. This mechanism resets membrane potential to zero when a spike occurs and remains inactive otherwise.

$$\mathbf{U}^l[t] = \tilde{\mathbf{U}}^l[t] \cdot \left(1 - \mathbf{S}^l[t]\right). \tag{3}$$

**Vanilla uniform quantization.** Quantization can be grouped into non-uniform and uniform quantization. Non-uniform quantization exhibits superior performance, but requires specialized hardware support. In contrast, uniform quantization maps weights to integer grids with equal size, simplifying both computational complexity and hardware implementation (Zhu et al. (2016)). In this study, we explore the uniform quantization in QP-SNNs. The vanilla uniform quantization for weights in layer $l$, i.e., $\mathbf{W}^l$, can be formulated as follows,

$$\mathbf{W}^l_{int} = \left\lceil \frac{s(b)}{2} \cdot \left(\text{clamp}(\mathbf{W}^l; -1, 1) + z\right) \right\rfloor, \tag{4}$$

where $\text{clamp}(\cdot)$ is a clipping operator, $\lceil \cdot \rfloor$ is a rounding operator, $z$ is the zero-point, $b$ is the bit width, and $s(b) = 2^b - 1$ is the number of integer grids. We set $z$ to 1 and explore bit widths $b$ of 8, 4, 2. Therefore, Eq.(4) maps $\mathbf{W}^l$ onto the unsigned integer grid $\{0, \cdots, 2^b - 1\}$. To reconstruct $\mathbf{W}^l$ through their quantized counterparts, the de-quantization is defined as,

$$\hat{\mathbf{W}}^l = 2 \cdot \frac{\mathbf{W}^l_{int}}{s(b)} - z. \tag{5}$$

Consequently, the general definition for the quantization used in the QP-SNN baseline is stated as,

$$\mathbf{W}^l \approx \hat{\mathbf{W}}^l = \frac{2}{s(b)} \left\lceil \frac{s(b)}{2} \cdot \left(\text{clamp}(\mathbf{W}^l; -1, 1) + z\right) \right\rfloor - z. \tag{6}$$

The vanilla quantization greatly reduces the baseline's storage and computation demands, but suffers from the limited weight precision. This constrains the model's representation capability, reducing performance. In the next section, we resolve this issue by effectively using the assigned bit-width.

**Structured pruning.** Pruning can be classified as unstructured and structured pruning. Unstructured pruning enables high sparsity and excellent performance but requires specialized design for hardware acceleration. In contrast, structured pruning preserves the model's structure and is highly compatible with existing hardware accelerators. Currently, the most advanced structured pruning method in SNN is presented by (Li et al. (2024)). They prune convolutional kernels according to the spiking channel activity (SCA) criterion. We use this criterion in our QP-SNN baseline for the following analysis and comparison. For the weight tensor $\mathbf{W}^l \in \mathbb{R}^{c_l \times c_{l-1} \times k \times k}$, the SCA-based criterion evaluates and prunes kernels based on the magnitude of membrane potential. The importance evaluation for the $f$-th kernel, i.e., $\mathbf{W}^{l,f} \in \mathbb{R}^{c_{l-1} \times k \times k}$, is defined as,

$$\text{Score}(\mathbf{W}^{l,f}) = \frac{1}{B \cdot T} \cdot \left(\sum_{b=1}^{B} \sum_{t=1}^{T} \left\| \tilde{\mathbf{U}}^{l,f}[t] \right\| \right), \tag{7}$$

where $B$ is the number of samples per mini-batch, $T$ is the time step, $\|\cdot\|$ is the L1-norm, and $\tilde{\mathbf{U}}^{l,f}[t]$ is the membrane potential of the $f$-th feature map. As shown in Eq.(7), the SCA-based criterion regards positive values in $\tilde{\mathbf{U}}^{l,f}$ as excitatory postsynaptic potentials and negative values as inhibitory postsynaptic potentials, thus removing kernels that contribute less to the membrane potential. By unifying the SCA-based criterion, the number of parameter and computation in baseline is further reduced. Noteworthy, the performance of the pruned QP-SNN baseline model relies strongly on the scoring function. Therefore, the idea $\text{Score}(\mathbf{W})^{l,f}$ should accurately identify the important kernels.

## 4 METHOD

To enhance the performance of the QP-SNN baseline, we analyze and resolve the underlying issues in quantization and pruning that cause performance reduction. In quantization, we reveal that the baseline suffers from limited representation capability due to inefficient bit-width utilization, and propose the weight rescaling strategy to use bit-width more effectively. In pruning, we propose a novel pruning criterion for the QP-SNN baseline to more accurately remove redundant kernels.

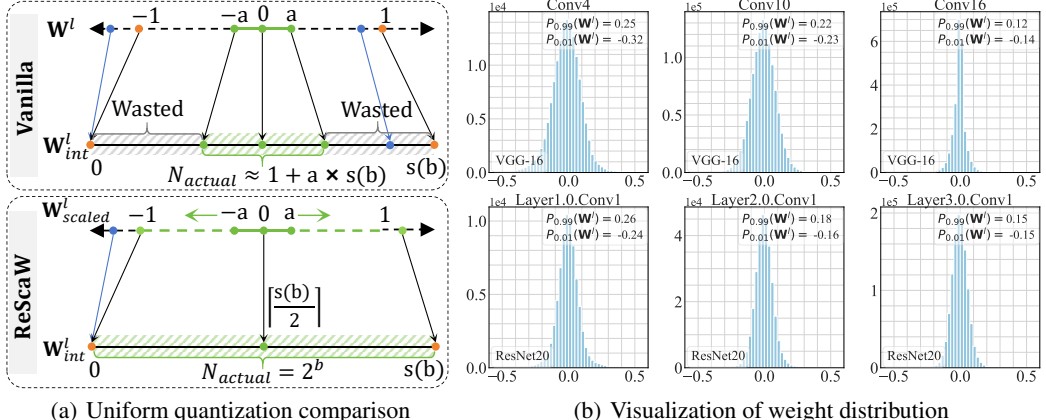

(a) Uniform quantization comparison         (b) Visualization of weight distribution

Figure 2: (a) Vanilla uniform quantization exhibits inefficient bit-width utilization, while ReScaW-based quantization can fully leverage the allocated bit-width. Green dots represent normal weights within the 1st and 99th percentiles, orange dots are boundary values, and blue dots are outliers. (b) Distribution is plotted to show that weights are concentrated in a narrow range around zero.

## 4.1 WEIGHT RESCALING STRATEGY

**Problem analysis.** The vanilla uniform quantization in the QP-SNN baseline minimizes resource usage, but suffers from inefficient bit-width utilization. This weakens the discrimination of the quantized weights in the QP-SNN baseline, limiting the model's representation capability. To evaluate the bit-width utilization efficiency, we define a metric as,

$$R_{utilize} = \frac{N_{actual}\left(\mathbf{W}_{int}^l\right)}{N_{total}\left(\mathbf{W}_{int}^l\right)}, \tag{8}$$

where $N_{actual}(\cdot)$, $N_{total}(\cdot)$ are the actual, available number of distinct values that $\mathbf{W}_{int}^l$ represents. Next, we analyze $R_{utilize}$ for QP-SNN baseline to assess bit-width utilization efficiency. We compute $N_{actual}(\mathbf{W}_{int}^l)$ using the range of full precision weights, shown in Figure 2(a)(top). We consider full precision weights between the 1st and 99th percentiles to eliminate outliers. For clarity, we denote $\mathbf{W}^l \in [-a, a]$, where $a = \max\left(\left|P_{0.01}(\mathbf{W}^l)\right|, \left|P_{0.99}(\mathbf{W}^l)\right|\right)$. Typically, $a$ is a positive value near zero (LeCun et al. (2002); He et al. (2015)). This means that the clamp function in Eq.(4) doesn't alter weight values. Based on this, we can deduce $\mathbf{W}_{int}^l \in \left[\left\lfloor\frac{z-a}{2}s(b)+\frac{1}{2}\right\rfloor, \left\lfloor\frac{z+a}{2}s(b)+\frac{1}{2}\right\rfloor\right]$. Therefore, $N_{actual}(\mathbf{W}_{int}^l)$ is approximately $(1+a \cdot s(b))$, leading to a utilization rate of $\frac{s(b)\cdot a+1}{s(b)+1}$.

To clearly show the low bit-width utilization of the baseline, we analyze the weight distribution and determine the value of $a$. We plot the weight distributions of VGG-16 and ResNet20 in Figure 2(b), and also label the 1st and 99th percentiles. In Figure 2(b), the smallest value of $a$ is 0.14 (VGG-16.conv16), and the largest is 0.32 (VGG-16.conv4). This indicates that under 8-bit quantization, the QP-SNN baseline uses less than half of the assigned bit width, with a minimum of 14.33% and a maximum of 32.26%. This inefficient utilization causes a large number of weights to be quantized to the same integer grid, reducing the discrimination of quantized weights. This limits the representation capacity of the QP-SNN baseline, leading to decreased performance. The complete weight distributions are provided in Appendix E.1.

**ReScaW-based uniform quantization.** To resolve the limited representation capacity, we propose a simple yet effective weight rescaling (ReScaW) strategy for the QP-SNN baseline that uses bit-width more efficiently. Specifically, we introduce a scale coefficient $\gamma$ to regulate the weight distribution to a wider range before quantization. The proposed ReScaW strategy is defined as,

$$\mathbf{W}_{scaled}^l = \frac{\mathbf{W}^l}{\gamma}. \tag{9}$$

The scaling coefficient $\gamma$ can assume any positive value within the range $0 < \gamma < 1$. We provide three options for $\gamma$: (1) the maximum absolute value: $max(|\mathbf{W}^l|)$; (2) the maximum absolute value of the

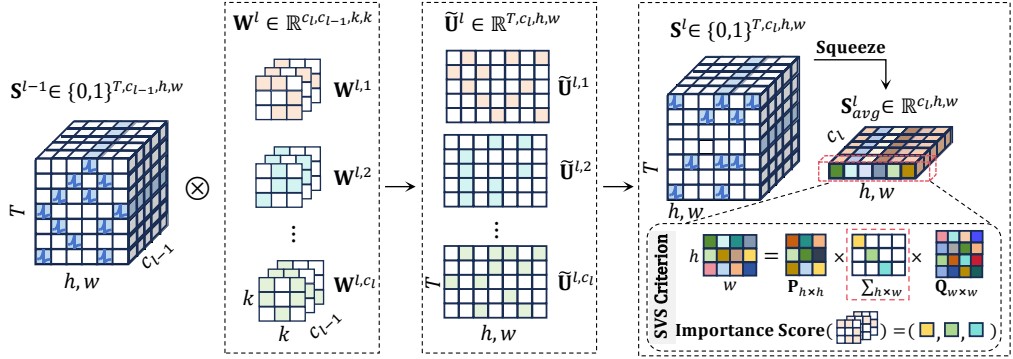

Figure 4: Proposed pruning criterion based on the singular value of spatiotemporal spike activity.

$x$-th and $(1-x)$-th percentiles: $\Psi_x(\mathbf{W}^l) = \max\left(\left|P_{1-x}(\mathbf{W}^l)\right|, \left|P_x(\mathbf{W}^l)\right|\right)$; and (3) 1-norm mean value: $\frac{\|\mathbf{W}^l\|_1}{|\mathbf{W}^l|}$, where $|\mathbf{W}^l|$ is the number of entries in $\mathbf{W}^l$. These three options can scale weights to span the range of $[-1, 1]$, thereby ensuring more efficient bit width utilization. The impact of these three options on performance will be explored in the experimental section. Consequently, we formulate the ReScaW-based uniform quantization as,

$$\mathbf{W}^l \approx \gamma \cdot \left(\frac{2}{s(b)}\left\lceil \frac{s(b)}{2} \cdot \left(\text{clamp}(\frac{\mathbf{W}^l}{\gamma}; -1, 1) + z\right)\right\rceil - z\right). \tag{10}$$

We compare vanilla uniform quantization in the QP-SNN baseline with the ReScaW in Figure 2(a). Clearly, the ReScaW method utilizes the allocated bit width more efficiently. This efficient bit-width utilization preserves the discrimination of quantized weights, enhancing the representation capability and performance of the QP-SNN baseline.

## 4.2 PRUNING CRITERION BASED ON THE SINGULAR VALUE OF SPIKE ACTIVITY

**Problem analysis.** The structured pruning work (Li et al. (2024)) employing the SCA-based criterion can produce high-performance pruned models, but the performance is ensured through multiple iterative pruning and regrowth processes. In fact, we observe that the SCA-based pruning criterion exhibits a high dependency on inputs. Specifically, it generates varying importance scores for the same convolutional kernel when processing different inputs. To prove this observation, we plot the importance scores of different kernels under varying inputs, as shown in Figure 3. The strong input dependency of SCA criterion can lead to biases in kernel evaluation, posing a risk of erroneously identifying crucial kernels as insignificant or misjudging unimportant kernels as essential. These misidentifications can affect the reliability of pruning, ultimately diminishing the performance of the QP-SNN baseline. Complete importance scores are available in Appendix F.1.

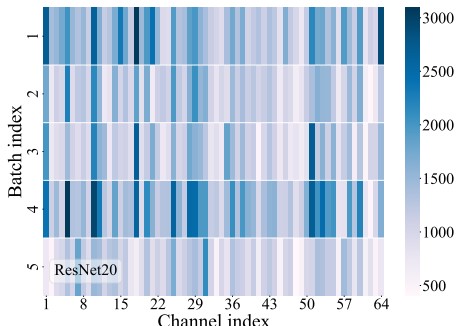

Figure 3: SCA assigns different scores to the same kernel for different input samples. The colors in the figure represent the value of importance score.

**SVS-based pruning criterion.** Several studies suggest that the number of singular values correlates with information richness (Sadek (2012); Baker (2005); Jaradat et al. (2021)). Inspired by this, we propose a novel pruning criterion for QP-SNN baseline using the singular value of spike activity (SVS) to remove redundant kernels precisely. As shown in Figure 4, the SVS-based criterion applies singular value decomposition to the average spike matrix over a given time window $T$, defined as,

$$\mathbf{S}_{avg}^{l,f} = \frac{1}{T}\sum_{t=1}^{T}\mathbf{S}^{l,f}[t] = \mathbf{P}\boldsymbol{\Sigma}_{h\times w}\mathbf{Q}^\top, \tag{11}$$

where $h$, $w$ are the height and width of the spike matrix. $\mathbf{P} = [\mathbf{p}_1, \ldots, \mathbf{p}_h] \in \mathbb{R}^{h \times h}$ and $\mathbf{Q} = [\mathbf{q}_1, \ldots, \mathbf{q}_w] \in \mathbb{R}^{w \times w}$ are orthogonal matrices representing the left and right singular vectors. $\mathbf{\Sigma} \in \mathbb{R}^{h \times w}$ is a diagonal matrix containing the singular values of $\mathbf{S}_{avg}^{l,f}$ in descending order, denoted as,

$$\mathbf{\Sigma} = \text{diag}\left(\sigma_1, \sigma_2, \cdots, \sigma_{min(h,w)}\right),$$
$$\text{with } \sigma_1 \geq \sigma_2 \geq \cdots \geq \sigma_{r^*} \geq \epsilon > \sigma_{r^*+1} \geq \cdots \geq \sigma_{min(h,w)} \geq 0. \tag{12}$$

Here, $\epsilon$ serves as a threshold to distinguish significant and negligible singular values, and it is generally set to a positive value near zero. Based on the threshold $\epsilon$, $\mathbf{S}_{avg}^{l,f}$ can be expressed as two components: $\mathbf{S}_{avg}^{l,f} = \sum_{i=1}^{r^*} \sigma_i \mathbf{p}_i \mathbf{q}_i^\top + \sum_{i=r^*+1}^{min(H,W)} \sigma_i \mathbf{p}_i \mathbf{q}_i^\top$. The first term captures the core feature dictated by the significant singular values, while the second term reflects potentially noise-related information (Jaradat et al. (2021)). Based on this decomposition, we define our pruning criterion as,

$$\text{Score}(\mathbf{W}^{l,f}) = \mathbb{E}_B \left( \sum_{i=1}^{min(H,W)} \mathbb{I}(\sigma_i > \epsilon) \right), \tag{13}$$

where $\mathbb{E}_B$ denotes the average over the mini-batch, and $\mathbb{I}(\cdot)$ is the indicator function that counts only significant singular values (those exceeding $\epsilon$) for importance evaluation. We will demonstrate in the experimental section that the proposed SVS-based pruning criterion achieves more accurate removal of unimportant convolutional kernels.

By integrating the ReScaW strategy and the SVS-based pruning criterion into the baseline, we develop QP-SNN, with its workflow outlined in Appendix A. In summary, the QP-SNN is lightweight and hardware-friendly, while also achieving high performance, making it an efficient solution for applications in resource-constrained scenarios.

## 5 EXPERIMENT

In this section, we first present the experiment setup. Second, we evaluate the performance of QP-SNN by comparing it to existing methods. Third, we demonstrate QP-SNN's scalability to complex architectures and tasks. Finally, we conduct extensive ablation studies to verify the effectiveness of the ReScaW and the SVS-based pruning criterion. Additional analyses of quantization-pruning order and QP-SNN's efficiency advantages are provided in Appendices B and C, respectively.

### 5.1 EXPERIMENT SETUP

We first evaluate our method on image classification tasks, including static datasets like CIFAR (Krizhevsky et al. (2009)), TinyImageNet, and ImageNet-1k (Deng et al. (2009)), alongside neuromorphic DVS-CIFAR10 (Li et al. (2017)). These datasets serve as standard benchmarks in machine learning and neuromorphic computing. For architecture, we use classical structures VGGNet and Spiking ResNet (Zheng et al. (2021)), with details provided in Table 1. We use SEW-ResNet (Fang et al. (2021)) on ImageNet-1k for a fair comparison with (Shi et al. (2024)). As for $\epsilon$ in Eq.(13), we observe minimal variation in the singular values, so we set it to a small value of 1e-6 (Jaradat et al. (2021)). For the learning of QP-SNN, we use the surrogate gradient and straight-through estimator to handle the nondifferentiability. We provide additional details in the appendix, with the learning algorithm described in Appendix D and experimental setups in Appendix G.

### 5.2 PERFORMANCE COMPARISON

As shown in Table 1, we compare QP-SNN with related work in performance and model size to prove the effectiveness and efficiency. We use 8, 4, and 2-bit weight configurations in experiments across all datasets. Compared to ANN2SNN conversion and hybrid algorithms, QP-SNN achieves top-1 performance with fewer timesteps, such as 2 or 4 on static datasets. When compared to direct algorithms, QP-SNN also performs well. On CIFAR-10 and CIFAR-100, QP-SNN outperforms previous methods (Shi et al. (2024); Li et al. (2024)) with smaller models and shorter timesteps (e.g., CIFAR-10: 1.61 MB, Acc=95.06%, T=2; CIFAR-100: 1.79 MB, Acc=75.13%, T=2). On TinyImageNet, using the same timesteps and architecture, QP-SNN reduces model size by 90.26% and increase accuracy by 3.71% compared to (Li et al. (2024)). On ImageNet, we are the first study to report results for structured pruning in SNNs, achieving comparable performance to unstructured

Table 1: Performance comparison on static and neuromorphic datasets. **Note:** 'H', 'D', and 'C' represent hybrid, direct, and conversion learning, respectively. 'HardF' denotes 'hardware-friendly'.

| Dataset | Method | Network | Train | Bits | HardF | Accuracy(%) | Timestep | Size (MB) |
|---------|--------|---------|-------|------|-------|-------------|----------|-----------|
| CIFAR-10 | Chowdhury et al. (2021) [*IJCNN21*] | VGG-9 | H | 32 | ✓ | 90.02 | 100 | 44.52 |
| | | | H | 5 | ✓ | 88.60 | 25 | 12.59 |
| | Deng et al. (2021) [*TNNLS21*] | 7Conv2FC | D | 32 | ✗ | 90.19 | 8 | 62.16 |
| | | | D | 3 | ✗ | 87.59 | 8 | 5.84 |
| | Shi et al. (2024) [*ICLR24*] | 6Conv2FC | D | 32 | ✗ | 92.63 | 8 | 50.28 |
| | | | D | 32 | ✗ | 90.65 | 8 | 28.40 |
| | Li et al. (2024) [*ICML24*] | VGG-16 | D | 32 | ✓ | 91.67 | 4 | 17.32 |
| | | | D | 32 | ✓ | 90.26 | 4 | 5.68 |
| | **Proposed QP-SNN** | ResNet20 | D | 8, 4, 2 | ✓ | 95.12, 95.41, 95.06 | 2 | 6.27, 3.16, 1.61 |
| | | | D | 8, 4, 2 | ✓ | 94.56, 94.65, 94.44 | 2 | 3.92, 1.98, 1.02 |
| | | VGG-16 | D | 8, 4, 2 | ✓ | 91.98, 91.90, 91.61 | 4 | 4.28, 2.16, 1.10 |
| | | | D | 8, 4, 2 | ✓ | 91.30, 91.19, 90.59 | 4 | 1.45, 0.74, 0.39 |
| CIFAR-100 | Chowdhury et al. (2021) [*IJCNN21*] | VGG-11 | H | 32 | ✓ | 67.80 | 50 | 75.90 |
| | | | H | 5 | ✓ | 66.20 | 30 | 25.43 |
| | Deng et al. (2021) [*TNNLS21*] | 7Conv2FC | D | 3 | ✗ | 57.83 | 8 | 11.75 |
| | | | D | 1 | ✗ | 55.95 | 8 | 5.99 |
| | Shi et al. (2024) [*ICLR24*] | ResNet18 | D | 32 | ✗ | 72.34 | 4 | 13.18 |
| | | | D | 32 | ✗ | 70.45 | 4 | 7.67 |
| | Li et al. (2024) [*ICML24*] | VGG-16 | D | 32 | ✓ | 65.53 | 4 | 14.40 |
| | | | D | 32 | ✓ | 64.64 | 4 | 9.48 |
| | **Proposed QP-SNN** | ResNet20 | D | 8, 4, 2 | ✓ | 75.29, 75.77, 75.13 | 2 | 6.45, 3.35, 1.79 |
| | | | D | 8, 4, 2 | ✓ | 74.78, 74.73, 73.89 | 2 | 4.10, 2.17, 1.20 |
| | | VGG-16 | D | 8, 4, 2 | ✓ | 66.69, 66.21, 65.69 | 4 | 2.48, 1.35, 0.79 |
| | | | D | 8, 4, 2 | ✓ | 64.70, 64.22, 63.08 | 4 | 1.85, 1.04, 0.63 |
| TinyImageNet | Kundu et al. (2021) [*WACV21*] | VGG-16 | C | 32 | ✗ | 52.70 | 150 | 24.21 |
| | Li et al. (2024) [*ICML24*] | VGG-16 | D | 32 | ✓ | 49.36 | 4 | 27.92 |
| | | | D | 32 | ✓ | 49.14 | 4 | 19.76 |
| | **Proposed QP-SNN** | VGG-16 | D | 8, 4, 2 | ✓ | 53.32, 53.11, 53.07 | 4 | 5.90, 3.78, 2.72 |
| | | | D | 8, 4, 2 | ✓ | 51.99, 51.78, 51.67 | 4 | 4.67, 3.17, 2.41 |
| ImageNet | Shi et al. (2024) [*ICLR24*] | ResNet18 | D | 32 | ✗ | 61.89 | 4 | 15.72 |
| | | | D | 32 | ✗ | 60.00 | 4 | 12.40 |
| | | | D | 32 | ✗ | 58.99 | 4 | 10.48 |
| | **Proposed QP-SNN** | ResNet18 | D | 8 | ✓ | 61.36 | 4 | 13.28 |
| | | | D | 4 | ✓ | 58.06 | 4 | 7.71 |
| DVS-CIFAR10 | Shi et al. (2024) [*ICLR24*] | VGGSNN | D | 32 | ✗ | 81.90 | 10 | 14.08 |
| | | | D | 32 | ✗ | 78.30 | 10 | 7.24 |
| | Li et al. (2024) [*ICML24*] | 5Conv1FC | D | 32 | ✓ | 73.00 | 20 | 3.92 |
| | | | D | 32 | ✓ | 71.90 | 20 | 0.32 |
| | **Proposed QP-SNN** | VGGSNN | D | 8, 4, 2 | ✓ | 82.10, 81.80, 81.30 | 10 | 1.61, 0.90, 0.55 |
| | | | D | 8, 4, 2 | ✓ | 81.50, 80.90, 80.50 | 10 | 1.05, 0.62, 0.41 |
| | | | D | 8, 4, 2 | ✓ | 75.90, 75.40, 74.90 | 10 | 0.40, 0.29, 0.24 |

pruning method (Shi et al. (2024)) with a 15.55% reduction in model size. On DVS-CIFAR10, QP-SNN achieves an 88.55% smaller model size and a 0.2% higher accuracy compared to (Shi et al. (2024)). To intuitively demonstrate QP-SNN's improvements, we plot comparison results in Figure 1. These results show that QP-SNN achieves superior results in both efficiency and performance, positioning it as a leading approach for compact and high-performance SNNs.

## 5.3 SCALABILITY TO COMPLEX ARCHITECTURES AND TASKS

QP-SNN can be extended to complex architectures like Transformer and complex tasks like object detection. To validate this scalability, we conduct two additional experiments: (1) using the Spikingformer (Zhou et al. (2023)) architecture and (2) applying QP-SNN to object detection task.

**Scalability to the Spikingformer.** We use the Spikingformer-4-384 architecture and evaluate it on the CIFAR-100 dataset. The training setups are consistent with (Zhou et al. (2023)). As shown in Table 2, QP-SNN reduces model size by $8.28\times$, SOPs by $2.25\times$, and power consumption by $2.25\times$, while achieving a competitive accuracy of 76.94%. These results demonstrate the effectiveness of QP-SNN in compressing complex Spiking Transformer architectures.

Table 2: Performance of QP-SNN on CIFAR-100 with the Spikingformer architecture.

| Architecture | Method | Connection | Bit | Model size (MB) | SOPs (M) | Power (mJ) | Accuracy (%) |
|---|---|---|---|---|---|---|---|
| Spikingformer | Zhou et al. (2023) | 100% | 16 | 18.64 base | 292.14 base | 0.266 base | 79.09 base |
| Spikingformer | QP-SNN | 44.74% | 4 | 2.25 8.28× | 130.05 2.25× | 0.118 2.25× | 76.94 -2.15 |

**Scalability to object detection.** We conduct object detection experiments on two remote sensing datasets: SSDD (Wang et al. (2019)) and NWPU VHR-10 (Cheng et al. (2017)). SSDD consists of synthetic aperture radar images for ship detection, while NWPU VHR-10 is a high-resolution remote sensing dataset containing ten object classes. We use the YOLO-v3 detection architecture with ResNet10 as the backbone. During training, we apply pruning to the backbone and optimize the model using SGD with a polynomial decay learning rate schedule, initializing at 1e-2 for 300 epochs. As shown in Table 3, QP-SNN significantly reduces model size while maintaining satisfactory detection performance, demonstrating its potential for more challenging tasks.

Table 3: Object detection results of QP-SNN on SSDD and NWPU VHR-10.

| Dataset | Method | Bit | Model size (MB) | mAP@0.5 (%) |
|---|---|---|---|---|
| SSDD | Full-precision | 32 | 19.29 base | 96.80 base |
| | QP-SNN | 4 | 2.15 8.97× | 97.10 +0.3 |
| NWPU VHR-10 | Full-precision | 32 | 19.29 base | 89.89 base |
| | QP-SNN | 4 | 2.15 8.97× | 86.68 -3.21 |

## 5.4 ABLATION STUDY

To prove the effectiveness of QP-SNN, we conduct extensive ablation studies. Firstly, we analyze the three options for $\gamma$ in the ReScaW strategy to select the optimal one. Then, we perform thorough ablation experiments to validate the effectiveness of the proposed ReScaW strategy and SVS-based pruning criterion. Finally, we visualize the effect of the ReScaW strategy and the SVS-based criterion to demonstrate that they have effectively addressed the above mentioned issues. All ablation experiments are conducted on the CIFAR-100 dataset using ResNet20 with 1.20 MB model size.

**Analysis of three options for $\gamma$.** We compare the performance of quantized SNNs (not involve pruning process) with different $\gamma$ settings to determine the optimal one as the default experimental setting. As depicted in Figure 5(a), the quantized SNN using $max(|\mathbf{W}^l|)$ achieves an accuracy of 77.85%, the one using $\Psi_x(\mathbf{W}^l)$ achieves an accuracy of 77.8%, and the one using $\left\|\mathbf{W}^l\right\|_1/|\mathbf{W}^l|$ achieves an accuracy of 79.16%. Clearly, the accuracy differences between them are minimal, with the 1-norm mean value performs best. This may be because $\left\|\mathbf{W}^l\right\|_1/|\mathbf{W}^l|$ can effectively capture the characteristics of the full precision distribution (Rastegari et al. (2016); Qin et al. (2020)). Therefore, we choose the 1-norm mean value as the default experimental setting.

**Effectiveness of two proposed methods.** As shown in Table 4, we conduct extensive ablation experiments to validate the effectiveness of two proposed methods in QP-SNN, i.e., the ReScaW strategy and the SVS-based pruning criterion. First, we demonstrate the effectiveness of the ReScaW strategy. *A* and *B* are models that apply vanilla and ReScaW quantization for SNN respectively, without involving pruning process. Their comparison shows that ReScaW-based quantization improves the performance of the quantized SNN by 0.63% over vanilla quantization, highlighting its effectiveness. Moreover, the comparison between Models *C* and *D* indicates that merely replacing the quantization in the baseline also results in a significant performance gain of 4.24%, further

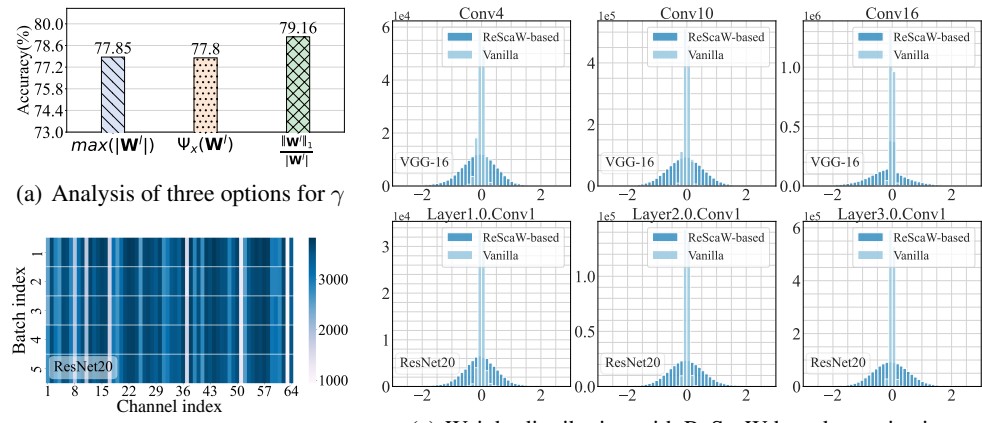

(a) Analysis of three options for $\gamma$

(b) Kernel scores with SVS criterion

(c) Weight distribution with ReScaW-based quantization

Figure 5: Visualization of ablation experiments.

Table 4: Ablation study on the effectiveness of two proposed methods. **Note:** 'Increment' represents the accuracy improvement relative to the specified model; ' r.w./ ' denotes 'replaced with'.

|  | Model | Accuracy(%) | Compared to | Increment(%) |
|---|---|---|---|---|
| A. | only vanilla quant | 78.53 | - | - |
| B. | only ReScaW quant | 79.16 | A. | 0.63 ↑ |
| C. | baseline | 69.16 | - | - |
| D. | r.w./ ReScaW | 73.40 | C. | 4.24 ↑ |
| E. | r.w./ SVS | 73.32 | C. | 4.16 ↑ |
| F. | r.w./ ReScaW & SVS | 73.89 | C. | 4.73 ↑ |

validating the ReScaW strategy. Second, we demonstrate the effectiveness of the SVS-based pruning criterion. The comparison between Models $C$ and $E$ reveals that the SVS criterion enhances the baseline performance by 4.16%, confirming its ability to remove kernels accurately and preserve model performance. By integrating these two methods into baseline, the performance is significantly improved by 4.73%, underscoring their importance for preserving QP-SNN performance.

**Impact of the ReScaW strategy and the SVS-based pruning criterion.** To demonstrate that the ReScaW strategy and SVS-based criterion effectively address the previously mentioned issues, we present the weight distribution and importance scores of QP-SNN. We depict the weight distribution of QP-SNN in Figure 5(c). This indicates that ReScaW-based quantization results in a broader weight distribution compared to vanilla quantization, indicating improved bit-width utilization efficiency. In addition, Figure 5(b) depicts the importance scores of QP-SNN, showing that the SVS-based pruning criterion produces stable scores with minimal fluctuation across different inputs. This input-insensitive characteristic enables QP-SNN to remove redundant kernels accurately. Complete visualization of weight distributions and kernel scores are provided in Appendix E.2 and F.2.

## 6 CONCLUSION

SNNs offer energy-efficient solutions for artificial intelligence. However, the current SNN community focuses mainly on building large-scale SNNs to increase performance, which limits their feasibility in resource-constrained edge devices. To tackle this limitation, we first developed a QP-SNN baseline using uniform quantization and structured pruning, which significantly reduces resource usage. Furthermore, we analyzed and addressed the underlying issues of the QP-SNN baseline in quantization and pruning to improve performance. For quantization, we revealed that the vanilla uniform quantization suffers from limited representation capability due to inefficient bit-width utilization and proposed a weight rescaling strategy to resolve it. For pruning, we observed that the SCA criterion exhibits low robustness on inputs and introduced a novel criterion using the singular value of spike activity to remove unimportant kernels more accurately. By integrating the ReScaW and SVS pruning criteria, our QP-SNN achieved superior efficiency and performance, demonstrating its potential for advancing neuromorphic intelligent systems and edge computing.

## ACKNOWLEDGMENTS

This work was supported in part by the National Natural Science Foundation of China under grant U20B2063, 62220106008, and 62106038, the Sichuan Science and Technology Program under Grant 2024NSFTD0034 and 2023YFG0259, the Open Research Fund of the State Key Laboratory of Brain-Machine Intelligence, Zhejiang University (Grant No.BMI2400020).

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

# A  THE OVERALL WORKFLOW OF QP-SNN

We present the workflow of QP-SNN in Algorithm 1, which consists of two main steps: quantization and pruning. First, the initial SNN model undergoes ReScaW-based uniform quantization, where weights are rescaled and quantized, followed by iterative training with backpropagation. Then, the quantized model is pruned using an SVS-based criterion that analyzes spatiotemporal spike activity to identify and remove less important channels. Finally, the quantized and pruned model is fine-tuned to restore performance.

---

**Algorithm 1:** The overall workflow of QP-SNN.

---

**Input:** Initial SNN model: $\mathcal{M} = \{\mathbf{W}^1, \cdots, \mathbf{W}^L\}$; Bit width: $b$; Pruning channel ratio: $r$; Number of training epoch: $N_{epoch}$; Number of training iteration per epoch: $I_{train}$.

**Output:** The trained QP-SNN $\mathcal{M}_{q\&p}$.

1 $\triangleright$ Step 1: Get quantized SNN $\mathcal{M}_q$ by using the ReScaW-based uniform quantization;

2 **for** $epoch \leftarrow 1$ **to** $N_{epoch}$ **do**

3     **for** $i \leftarrow 1$ **to** $I_{train}$ **do**

4         **for** $l \leftarrow 1$ **to** $L$ **do**

5             $\gamma \in \{\max(|\mathbf{W}^l|), \Psi_x(\mathbf{W}^l), \frac{\|\mathbf{W}^l\|_1}{|\mathbf{W}^l|}\}$;       $\triangleright$ The selection is fixed during training;

6             $\mathbf{W}^l_{scaled} = \frac{\mathbf{W}^l}{\gamma}$;       $\triangleright$ Rescale 32-bit weight parameters to a wide range;

7             $\mathbf{W}^l \approx \gamma \cdot \left(\frac{2}{s(b)} \left\lceil \frac{s(b)}{2} \cdot \left(\text{clamp}(\frac{\mathbf{W}^l}{\gamma}; -1, 1) + z\right) \right\rceil - z\right)$;

8             **for** $t \leftarrow 1$ **to** $T$ **do**

9                 Calculate $\tilde{\mathbf{U}}^l[t]$, $\mathbf{S}^l[t]$, and $\mathbf{U}^l[t]$ according to Eq.(1~3)

10             **end**

11         **end**

12         Perform backpropagation and update the quantized model parameters $\mathcal{M}_q$;

13     **end**

14 **end**

15 $\triangleright$ Step 2: Get the pruned QP-SNN $\mathcal{M}_{q\&p}$ with the SVS-based pruning criterion;

16 **for** $l \leftarrow 1$ **to** $L$ **do**

17     Initialize an array $\mathcal{F}$;

18     **for** $f \leftarrow 1$ **to** $n_l$ **do**

19         Perform a inference process with mini-batch data, get spatiotemporal spike activity;

20         Get the singular value matrix $\mathbf{\Sigma}$: $\frac{1}{T} \sum_{t=1}^{T} \mathbf{S}^{l,f}[t] = \mathbf{P}\mathbf{\Sigma}_{h \times w}\mathbf{Q}^{\top}$;

21         $\text{Score}(\mathbf{W}^{l,f}) = \mathbb{E}_B \left(\sum_{i=1}^{min(H,W)} \mathbb{I}(\sigma_i > \epsilon)\right)$;

22         $\mathcal{F}[f] = \text{Score}(\mathbf{W}^{l,f})$;

23     **end**

24     $I_{prun} = \lceil r * n_l \rceil$; $s_{index} = argsort(\mathcal{F})[I_{prun} :]$;       $\triangleright$ Select kernels with high score;

25     $\mathbf{W}^l_{q\&p} \leftarrow$ Kernels with index in $s_{index}$;

26 **end**

27 **function** $Finetune(\mathcal{M}_{q\&p})$;       $\triangleright$ Fine-tune the pruned model to optimize performance.

---

# B  ANALYSIS ON THE ORDER OF QUANTIZATION AND PRUNING

When two or more model lightweight techniques are employed, compatibility issues often arise, such as the order of applying these techniques and the training strategies involved. In this paper, we adopt the 'quantize first, then prune' strategy based on the following two considerations. **First**, this strategy can better guarantee the effect of pruning technique. Specifically, if pruning is applied before quantization, important convolutional kernels identified in the full-precision parameter domain may become misaligned after quantization, as the quantization reintroduces additional errors. In contrast, by quantizing first and then pruning, redundant convolutional kernels are identified directly in the target low-precision parameter domain. This order allows for more accurate identification and preservation of critical kernels. **Second**, this strategy significantly reduces training overhead. Pruning before quantization requires three weight updates: 'full-precision SNN training→ pruning with fine-tuning → quantization with fine-tuning,' while 'quantize first, then prune' only requires two adjustments: 'quatized SNN training→ pruning with fine-tuning.'

In addition to the theoretical analysis, we have also conducted experiments by reversing the order of quantization and pruning, termed PQ-SNN, to validate the effectiveness of QP-SNN. Experiments are performed on the CIFAR-100 with ResNet20 under the bit-width of 4. We summarize the experimental results in Table 5, from which two conclusions can be obtained. **First**, the proposed ReScaW and SVS can improve both performance, regardless of the order in which they are applied, leading to a 1.83% improvement in PQ-SNN and a 4.46% improvement in QP-SNN. This proves the effectiveness of our ReScaW and SVS methods. **Second**, QP-SNN achieves the highest performance (surpassing PQ-SNN by 1.39%), demonstrating that 'quantize first, then prune' is more effective.

Table 5: Ablation study on the order of quantization and pruning.

| Method | PQ-SNN baseline | PQ-SNN | QP-SNN baseline | QP-SNN |
|---|---|---|---|---|
| Accuracy | 71.51% | 73.34%$_{(baseline+1.83\%)}$ | 70.27% | 74.73%$_{(baseline+4.46\%)}$ |

## C  EFFICIENCY VALIDATION OF QP-SNN

Model compression aims to optimize efficiency during the inference phase, facilitating efficient deployment on resource-constrained devices. Therefore, we present the key efficiency metrics of QP-SNN during inference, including model size, SOPs, power consumption, and accuracy, to verify the efficiency advantage of QP-SNN.

We first present a comparison of our model with the full-precision uncompressed SNN counterparts. The results are summarized in Table 6. We acknowledge that our method exhibits accuracy loss compared to uncompressed SNNs. However, this performance degradation is a common challenge in the field of model compression. Fortunately, QP-SNN demonstrates satisfactory performance under extreme compression ratios. For example, on the CIFAR-10 dataset, under the extreme connection ratio of 9.61%, QP-SNN reduces the model size by 98.74%, SOPs by 78.69%, and power consumption by 77.45%, while the accuracy decreases by only 2.44%. This trade-off between performance degradation and resource efficiency is highly advantageous in edge computing scenarios.

Table 6: Efficiency metrics comparison of QP-SNN with full-precision uncompressed SNN.

|  | Architecture | Connection | Bit | Model size (MB) | SOPs (M) | Power (mJ) | Accuracy |
|---|---|---|---|---|---|---|---|
| CIFAR-10 | VGG-16 | 100% | 32 | 58.88 | 54.60 | 0.204 | 93.63% |
|  | VGG-16 | 9.61% | 4 | 0.74 | 11.63 | 0.046 | 91.19% |
| CIFAR-100 | ResNet20 | 100% | 32 | 68.4 | 415.64 | 0.756 | 79.49% |
|  | ResNet20 | 22.69% | 4 | 2.17 | 131.53 | 0.126 | 74.73% |

We then add a comparison of our method with related studies on CIFAR-10. Experimental results are shown in Table 7. It can be seen that QP-SNN exhibits competitive SOPs compared to compression work in the SNN domain, and exhibits extremely low model size due to quantization. Moreover, it is worth noting that the advanced works (Deng et al. (2021); Shi et al. (2024)) focus on unstructured pruning, which typically achieves higher sparsity and performance but requires specialized hardware support. In contrast, our work adopts uniform quantization and structured pruning, balancing the advantages of sparsity, performance, and hardware compatibility.

Table 7: Efficiency metrics comparison of QP-SNN with related studies on the CIFAR-10 dataset.

| Method | Architecture | Time step | HardF | Model size (MB) | SOPs (M) |
|---|---|---|---|---|---|
| Deng et al. (2021) [TNNLS21] | 7Conv2FC | 8 | ✗ | 62.16 | 107.97 |
| Shi et al. (2024) [ICLR24] | 6Conv2FC | 8 | ✗ | 33.76 | 11.98 |
| Li et al. (2024) [ICML24] | VGG-16 | 4 | ✓ | 5.68 | - |
| QP-SNN | VGG-16 | 4 | ✓ | 0.74 | 11.63 |

# D  LEARNING ALGORITHM FOR QP-SNN

In this section, we introduce the learning algorithm for the QP-SNN. We use the spatio-temporal backpropagation (STBP) (Wu et al. (2018)) and the straight-through estimator (STE) (Bengio et al. (2013)) methods to solve the non-differentiability of the spike generation function and quantization.

Training QP-SNNs requires calculating the gradient of the loss function with respect to the synaptic weight. In this work, we use the STBP learning algorithm, which performs gradient propagation in both spatial and temporal dimensions. By applying the chain rule, STBP computes the derivative of the loss function $\mathcal{L}$ with respect to synaptic weights $\mathbf{W}^l$ through the following equation,

$$
\begin{aligned}
\frac{\partial \mathcal{L}}{\partial \mathbf{W}^l} = \sum_{t=1}^{T} & \frac{\partial \mathcal{L}}{\partial \mathbf{S}^{l+1}[t]} \frac{\partial \mathbf{S}^{l+1}[t]}{\partial \mathbf{U}^{l+1}[t]} \left( \frac{\partial \mathbf{U}^{l+1}[t]}{\partial \mathbf{W}^l} + \right. \\
& \left. \sum_{\tau < t} \prod_{i=t-1}^{\tau} \left( \frac{\partial \mathbf{U}^{l+1}[i+1]}{\partial \mathbf{U}^{l+1}[i]} + \frac{\partial \mathbf{U}^{l+1}[i+1]}{\partial \mathbf{S}^{l+1}[i]} \frac{\partial \mathbf{S}^{l+1}[i]}{\partial \mathbf{U}^{l+1}[i]} \right) \frac{\partial \mathbf{U}^{l+1}[\tau]}{\partial \mathbf{W}^l} \right),
\end{aligned}
\tag{14}
$$

where the derivative of the loss function with respect to the spike $\frac{\partial \mathcal{L}}{\partial \mathbf{S}^{l+1}[t]}$ is obtained in an iterative manner. The terms of $\frac{\partial \mathbf{U}^{l+1}[t]}{\partial \mathbf{W}^l}$, $\frac{\partial \mathbf{U}^{l+1}[i+1]}{\partial \mathbf{U}^{l+1}[i]}$ and $\frac{\partial \mathbf{U}^{l+1}[i+1]}{\partial \mathbf{S}^{l+1}[i]}$ can be computed from Eq.(1). Unfortunately, the direct training of SNNs faces a distinct challenge due to the non-differentiable nature of the spiking (i.e. firing) mechanism. Specifically, the term of $\frac{\partial \mathbf{S}^{l+1}[t]}{\partial \mathbf{U}^{l+1}[t]}$ represents the gradient of the spike generation function (described in Eq. (2)). This function evaluates to infinity at the moment of spike emission and to zero elsewhere, making it incompatible with the traditional error backpropagation used in ANN training. STBP addresses this non-differentiability problem by employing surrogate gradients to approximate the true gradient Wu et al. (2018). In this paper, we use the triangular surrogate function (Deng et al. (2022)), described as $\frac{\partial \mathbf{S}^{l+1}[t]}{\partial \mathbf{U}^{l+1}[t]} = \frac{1}{a} \max\left( a - |\mathbf{U}^{l+1}[t] - \theta|, 0 \right)$, where $a$ is the coefficient that controls the width of the gradient window. In this paper, we use the cross-entropy loss function to access the difference between the predicted probability distribution and the true label, given by, $\mathcal{L} = -\sum_{i=1}^{N_L} y_i \log\left( \frac{exp(\frac{1}{T}\sum_{t=1}^{T} \tilde{\mathbf{U}}_i^L[t])}{\sum_j exp(\frac{1}{T}\sum_{t=1}^{T} \tilde{\mathbf{U}}_j^L[t])} \right)$, where $N_L$ is the number of classes and $y_i \in \{0, 1\}$ is the label for the $i$-th neuron in the last layer. Moreover, to solve the non-differentiability of quantization, we use the STE method (Hinton et al. (2012); Bengio et al. (2013)), expressed as, $\frac{\partial \mathbf{W}^l}{\partial \mathbf{W}_{int}^l} = 1_{|\mathbf{W}^l| \leq 1}$. By using the surrogate gradient function and STE, the proposed QP-SNN can be trained directly with gradient backpropagation.

# E  COMPLETE WEIGHT DISTRIBUTION COMPARISON

## E.1  VANILLA UNIFORM QUANTIZATION

We present the weight distributions of models utilizing vanilla uniform quantization across multiple datasets and architectures, such as ResNet20 on CIFAR100, VGG-16 on TinyImageNet, and VG-GSNN on DVS-CIFAR10. In addition to the weight distribution, we also label the 1st and 99th percentiles of each layer's weights in the figure to determine the value of $a$. Based on the value of $a$ and the utilization rate equation $\frac{s(b) \cdot a + 1}{s(b) + 1}$ in Sec. 4.1, we calculate the bit-width utilization for each model. In these calculations, we consider an 8-bit weight configuration, i.e., $s(b) = 256$.

The weight distribution of ResNet20 on the CIFAR-100 dataset is presented in Figure 6. It can be seen from this figure that only the weight distribution of the first layer is relatively wide, with an $a$ value of 0.7, which corresponds to a bit-width utilization rate of 70.12%. In contrast, the $a$ value for the subsequent layers are predominantly around 0.2, resulting in a significantly lower bit-width utilization rate of approximately 20.31%.

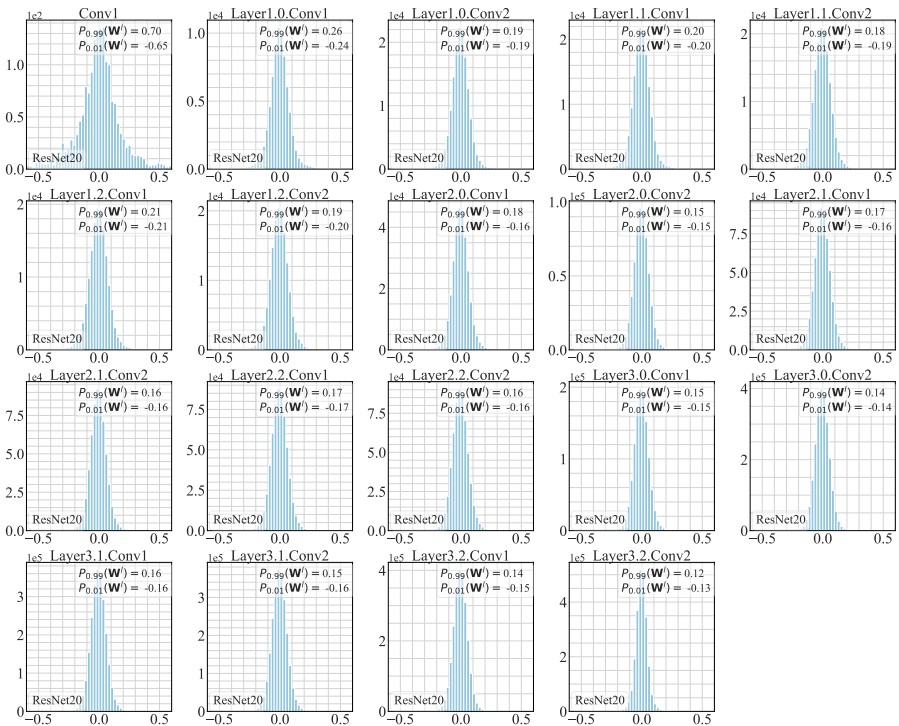

Figure 6: Weight distribution: vanilla uniform quantization, ResNet20, CIFAR-100.

The weight distribution of VGG-16 on TinyImagenet is illustrated in Figure 7. From this figure, it can be revealed that the weight distribution of each layer is broader than ResNet20 on CIFAR-100. However, the maximum $a$ value is 0.64, which corresponds to a bit-width utilization rate of approximately 64.14%. This result indicates it is still quite far from full utilization.

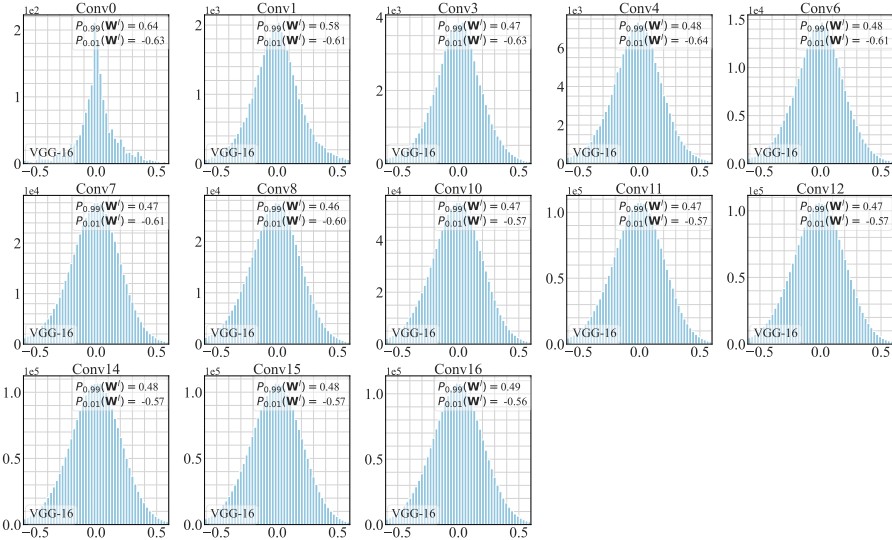

Figure 7: Weight distribution: vanilla quantization, VGG-16, TinyImageNet.

The weight distribution of VGGSNN on DVS-CIFAR10 is displayed in Figure 8. As can be seen from the figure, the maximum $a$ value is 0.44 (Conv1), corresponding to a bit width utilization rate of 44.22%. Moreover, the $a$ value of subsequent layers is mainly around 0.3, resulting in a significantly lower bit width utilization rate of about 30.27%.

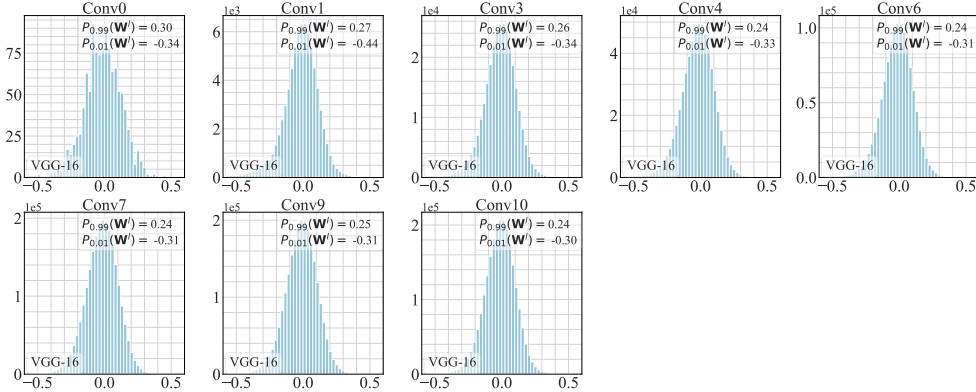

Figure 8: Weight distribution: vanilla quantization, VGGSNN, DVS-CIFAR10.

Clearly, these weight distributions prove the inefficient bit-width utilization of vanilla uniform quantization. This inefficiency leads to a substantial number of floating-point weights being discretized on the same integer grid during the quantization process, thus reducing the discrimination of the quantized weights. Consequently, this reduction adversely impacts the network's representational capacity and overall performance.

### E.2 ReScaW-based uniform quantization

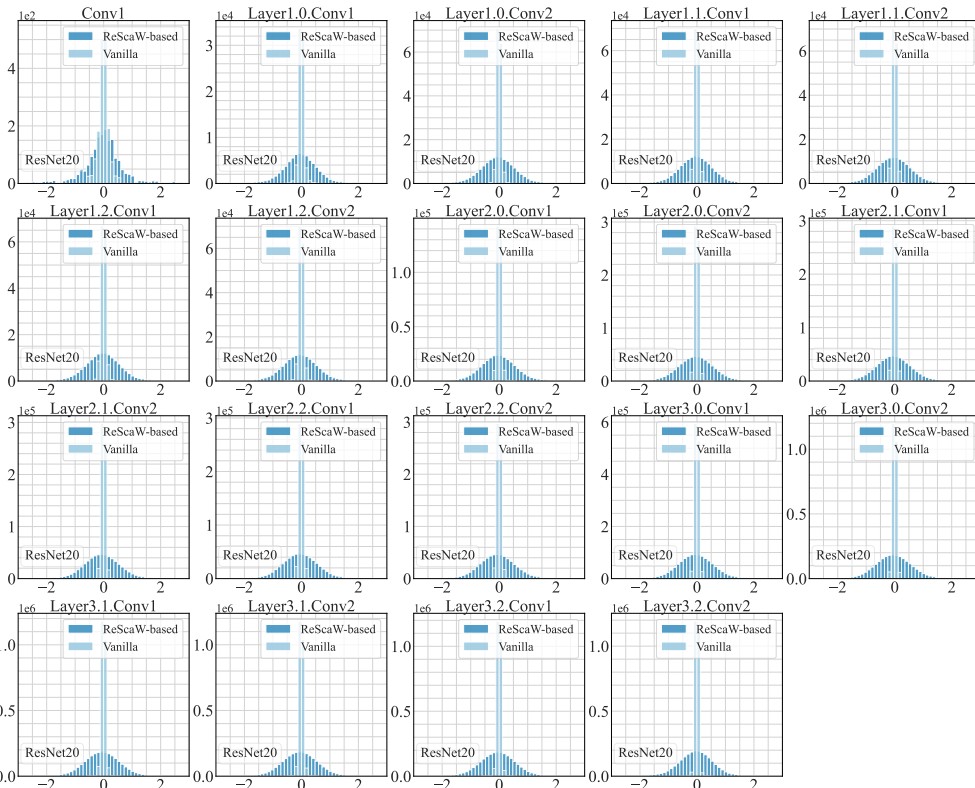

Figure 9: Weight distribution: ReScaW-based uniform quantization, ResNet20, CIFAR-100.

We also present a comparison of weight distributions between vanilla uniform quantization and ReScaW-based uniform quantization across multiple datasets and architectures. The weight distribution of ResNet20 on CIFAR-100 is presented in Figure 9, VGG-16 on TinyImageNet is illustrated

in Figure 10, and VGG-SNN on DVS-CIFAR10 is displayed in Figure 11. These three figures clearly demonstrate that the weight distribution using ReScaW-based quantization is broader than that of vanilla uniform quantization, indicating the more efficient bit-width utilization of our ReScaW.

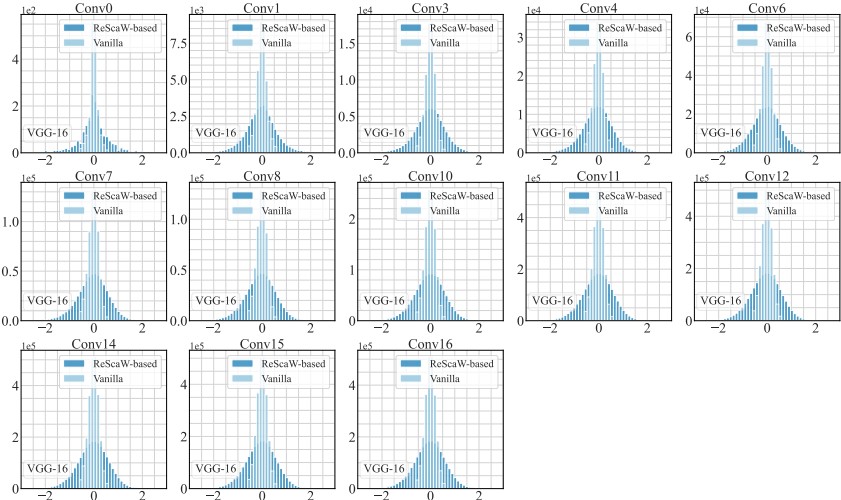

Figure 10: Weight distribution: ReScaW-based uniform quantization, VGG-16, TinyImageNet.

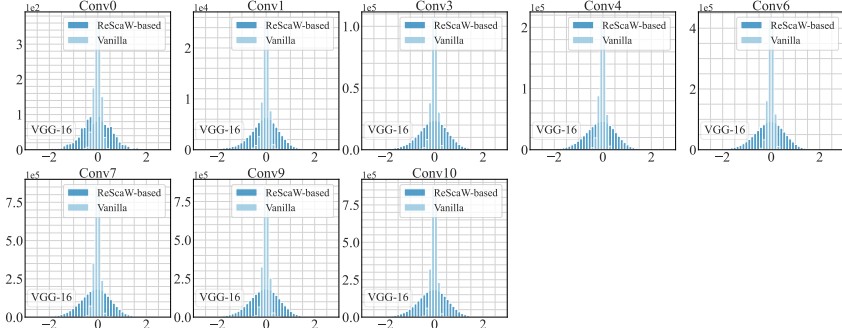

Figure 11: Weight distribution: ReScaW-based uniform quantization, VGGSNN, DVS-CIFAR10.

## F  COMPLETE IMPORTANCE SCORE COMPARISON

### F.1  SCA-BASED PRUNING CRITERION

We present the convolutional kernel scores of models using the SCA criterion across different architectures and datasets, including ResNet20 on CIFAR100, VGG-16 on TinyImageNet, and VGGSNN on DVS-CIFAR10. Note that we only display the layers that perform pruning operations, and the colors in these figures represent the value of the importance score. Moreover, to intuitively reflect the robustness of the pruning criterion to input samples, we compute the average cosine similarity of kernel scores between pairs of input batches for each layer in every model. The calculation for the average cosine similarity of $l$-th layer is outlined as,

$$\mathrm{AvgCosS}_l = \frac{2}{N_B(N_B - 1)} \sum_{i<j} \frac{\sum_f \mathrm{Score}_i(\mathbf{W}^{l,f}) \cdot \mathrm{Score}_j(\mathbf{W}^{l,f})}{\sqrt{\sum_f \mathrm{Score}_i(\mathbf{W}^{l,f})^2} \cdot \sqrt{\sum_f \mathrm{Score}_j(\mathbf{W}^{l,f})^2}} \tag{15}$$

where $N_B$ is the number of input batches and $\mathrm{Score}_i$ is the kernel score for input batch $i$.

The kernel scores for ResNet20 on CIFAR-100 are presented in Figure 12. It can be seen from this figure that the SCA-based pruning criterion yields varying scores for the same kernel when processing different input samples. Furthermore, we calculated $\mathrm{AvgCosS}_l$ for each layer in ResNet20, and the $\min_l \mathrm{AvgCosS}_l$ is 0.870. This indicates that the SCA criterion is not robust enough to inputs.

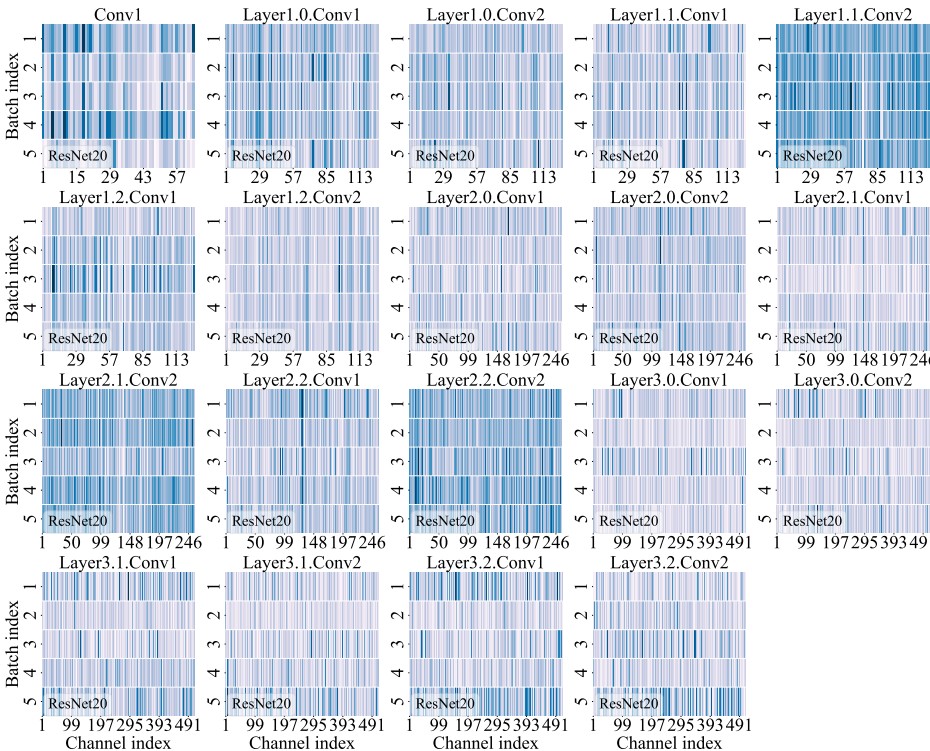

Figure 12: Kernel scores: SCA-based pruning criterion, ResNet20, CIFAR-100.

The kernel scores for VGG-16 on TinyImagenet are illustrated in Figure 13. We calculate $\text{AvgCosS}_l$ for each layer in VGG-16, and obtain the $\min_l \text{AvgCosS}_l$ is 0.879. In this structure, the kernel scores' fluctuation with inputs is slightly better compared to ResNet20, but still not negligible.

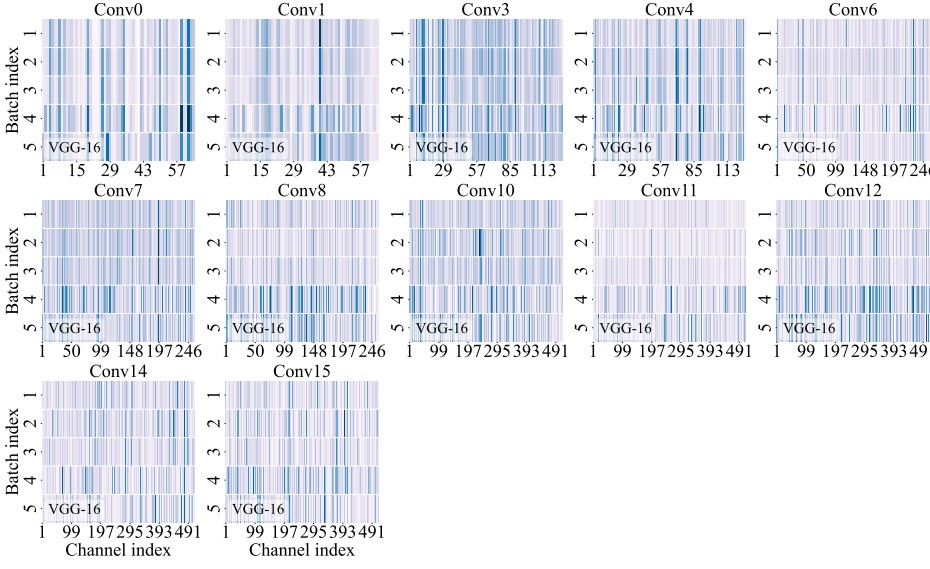

Figure 13: Kernel scores: SCA-based pruning criterion, VGG-16, TinyImageNet.

The kernel scores for VGGSNN on DVS-CIFAR-10 are displayed in Figure 14. As can be seen from the figure, The kernel score's fluctuation with input data is better compared to both ResNet and VGG-16, but in deeper layers, the fluctuation is higher. We also calculate $\text{AvgCosS}_l$ for each layer in VGGSNN, and the $\min_l \text{AvgCosS}_l$ is 0.952.

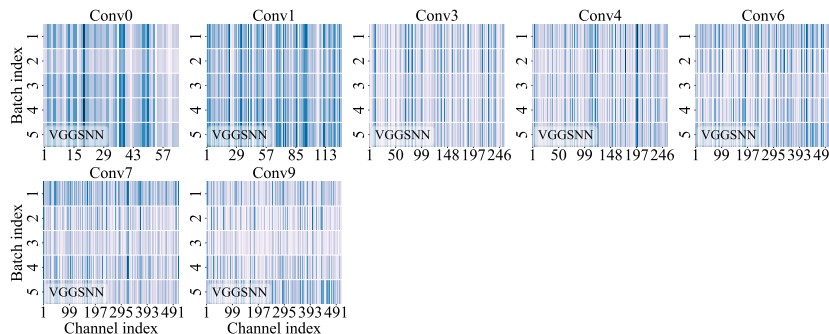

Figure 14: Kernel scores: SCA-based pruning criterion, VGGSNN, DVS-CIFAR10.

These results demonstrate that the SCA-based pruning criterion yields varying scores for the same kernel when processing different input sample, demonstrating low robustness to input samples. This sensitivity to inputs suggests that the criterion may fail to accurately identify critical convolutional kernels within SNNs, potentially impacting the reliability of the pruning process.

## F.2 SVS-BASED PRUNING CRITERION

We also depict kernel scores using the SVS pruning criterion. The kernel score for ResNet20 on CIFAR-100 in Figure 15, VGG-16 on TinyImagenet in Figure 16, and VGGSNN on DVS-CIFAR10 in Figure 17. We still only display the layers that perform pruning operation. In VGG-16, ResNet20, and VGGSNN, the $\min_l \text{AvgCosS}_l$ values are 0.997, 0.993, and 1.000 respectively, which exceed the corresponding $\min_l \text{AvgCosS}_l$ when using the SCA-Based pruning criterion by 13.4%, 14.1%, and 5.0%, respectively. The results demonstrate that the SVS-based pruning criterion yields consistent evaluations, with only minor variations between different input samples. This high robustness to input samples enables QP-SNN to effectively identify and eliminate redundant kernels.

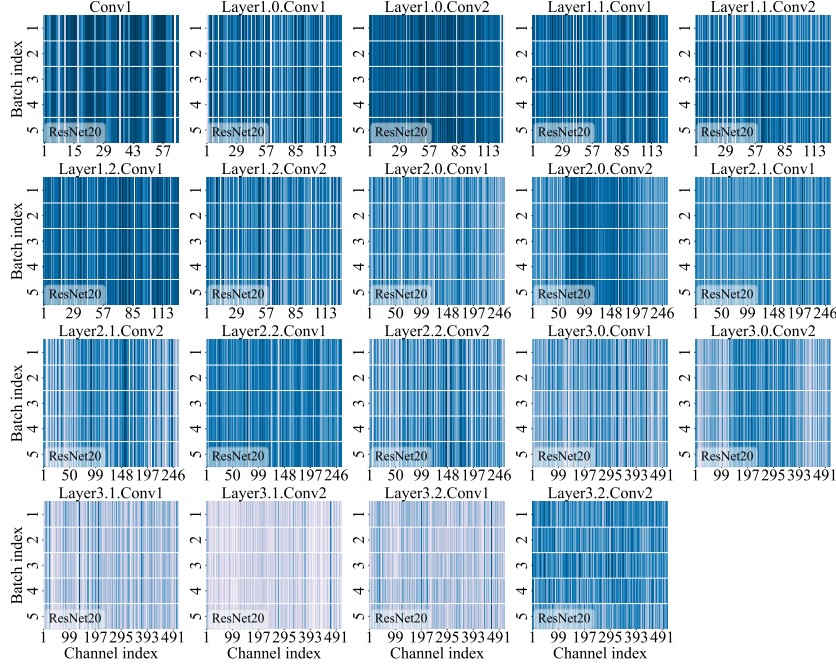

Figure 15: Kernel scores: SVS-based pruning criterion, ResNet20, CIFAR-100.

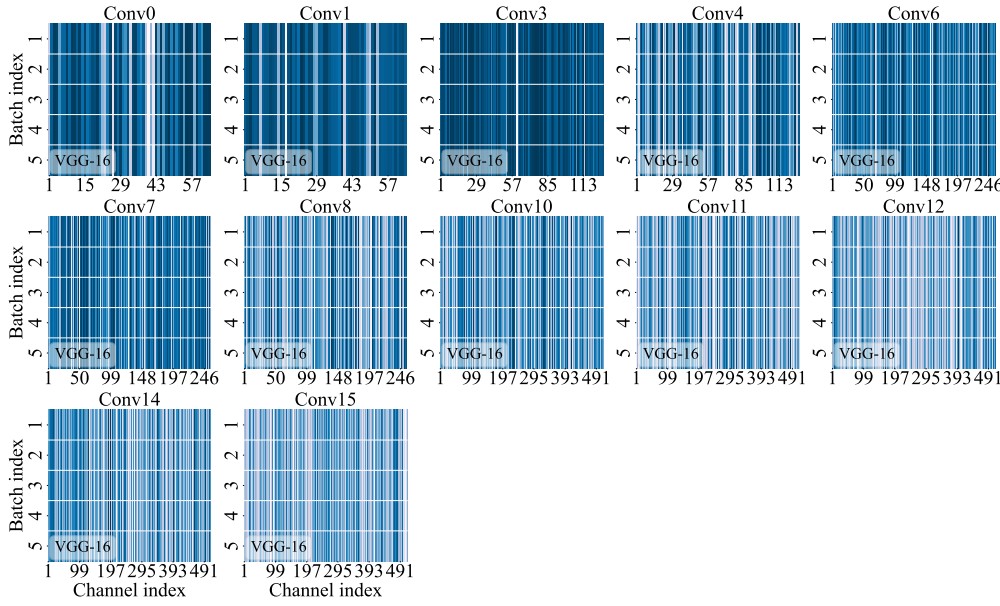

Figure 16: Kernel scores: SVS-based pruning criterion, VGG-16, TinyImageNet.

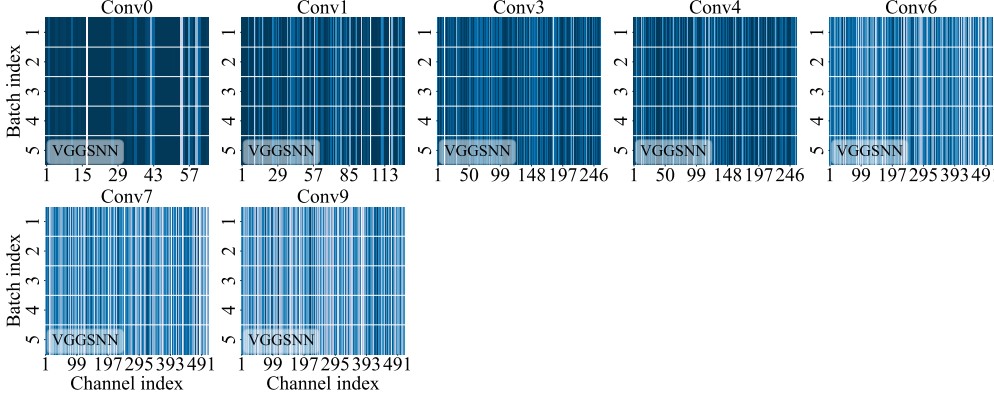

Figure 17: Kernel scores: SVS-based pruning criterion, VGGSNN, DVS-CIFAR10.

## G EXPERIMENT

**Datasets** We evaluate our method on image classification datasets, including static datasets CIFAR-10 (Krizhevsky et al. (2009)), CIFAR-100 (Krizhevsky et al. (2009)), TinyImageNet (Deng et al. (2009)), ImageNet-1k (Deng et al. (2009)), and the neuromorphic dataset DVS-CIFAR10 Li et al. (2017). Before introducing the experiment setups, we briefly outline each dataset. The CIFAR-10 and CIFAR-100 are color image datasets, with each dataset containing 50,000 training images and 10,000 testing images. Each image features 3 color channels and a spatial resolution of 32×32 pixels. CIFAR-10 is composed of 10 categories, whereas CIFAR-100 comprises 100 categories. During the preprocessing process of CIFAR datasets, we apply the commonly used data augmentation techniques (Cubuk et al. (2018); DeVries (2017)). The TinyImageNet dataset is a subset of the ImageNet dataset, consisting of 200 categories, with each category containing 500 training images and 50 test images. Each image has 3 color channels and a spatial resolution of 64×64 pixels. The ImageNet-1K dataset is a large-scale dataset commonly used for computer vision tasks. It spans 1000 classes and contains around 1.3 million training images and 50,000 validation images. The DVS-CIFAR10 is a neuromorphic dataset captured using Dynamic Vision Sensor (DVS) event cameras. It is the most challenging neuromorphic dataset, featuring 9,000 training samples and 1,000 testing samples, featuring a spatial resolution of 128×128. During the preprocessing process of the DVS-CIFAR10 dataset, we apply the data augmentation technique proposed in (Li et al. (2022)).

**Experimental Setups**   We summarize the training hyperparameters for each dataset in Table 8, including time step, image resolution, optimizer, and other factors. Additionally, we present the network architectures and the corresponding pruning rates for each module in Table (9~11). In our experiments, we directly utilize the classification head after completing the convolution operations. Therefore, we do not prune the output channels of the last convolutional layer to preserve the integrity of the classification head. Note that the pruning rates used in our experiments are manually selected, without rigorous design or the application of parameter search methods.

Table 8: Experimental setups.

| Hyper-parameter | CIFAR-10/100 | TinyImageNet | ImageNet | DVS-CIFAR10 |
|---|---|---|---|---|
| Timestep | 2, 4 | 4 | 4 | 10 |
| Resolution | 32×32 | 64×64 | 224×224 | 48×48 |
| Batch size | 256 | 256 | 256 | 64 |
| Epoch (Train/Fine-tune) | 300 / 150 | 300 / 150 | 320 / 200 | 300 / 150 |
| Optimizer (Train/Fine-tune) | SGD / Adam | SGD / Adam | SGD / SGD | SGD / Adam |
| Initial lr (Train/Fine-tune) | 0.1 / 0.001 | 0.1 / 0.001 | 0.1 / 0.05 | 0.1 / 0.001 |
| Learning rate decay | Cosine | Cosine | Cosine | Cosine |

Table 9: Detailed network architecture and the channel pruning ratio for VGG-16.

| Layer | Resolution | Channel | Module | Channel Pruning Ratio | | | | | |
|---|---|---|---|---|---|---|---|---|---|
| | | | | CIFAR-10 | | CIFAR-100 | | TinyImageNet | |
| | | | | 4.25M | 1.42M | 2.31M | 1.68M | 4.65M | 3.43M |
| 1 | $H \times W$ | 64 | Conv-BN-LIF | 0.45 | 0.49 | 0.45 | 0.45 | 0.45 | 0.45 |
| 2 | $H \times W$ | 64 | QConv-BN-LIF | 0.45 | 0.49 | 0.45 | 0.45 | 0.45 | 0.45 |
| 3 | - | - | MaxPool | - | | | | | |
| 4 | $\frac{H}{2} \times \frac{W}{2}$ | 128 | QConv-BN-LIF | 0.45 | 0.49 | 0.45 | 0.45 | 0.45 | 0.45 |
| 5 | $\frac{H}{2} \times \frac{W}{2}$ | 128 | QConv-BN-LIF | 0.45 | 0.49 | 0.45 | 0.45 | 0.45 | 0.45 |
| 6 | - | - | MaxPool | - | | | | | |
| 7 | $\frac{H}{4} \times \frac{W}{4}$ | 256 | QConv-BN-LIF | 0.45 | 0.49 | 0.45 | 0.45 | 0.45 | 0.45 |
| 8 | $\frac{H}{4} \times \frac{W}{4}$ | 256 | QConv-BN-LIF | 0.45 | 0.49 | 0.45 | 0.45 | 0.45 | 0.45 |
| 9 | $\frac{H}{4} \times \frac{W}{4}$ | 256 | QConv-BN-LIF | 0.45 | 0.49 | 0.45 | 0.45 | 0.45 | 0.45 |
| 10 | - | - | MaxPool | - | | | | | |
| 11 | $\frac{H}{8} \times \frac{W}{8}$ | 512 | QConv-BN-LIF | 0.51 | 0.8 | 0.7 | 0.78 | 0.51 | 0.62 |
| 12 | $\frac{H}{8} \times \frac{W}{8}$ | 512 | QConv-BN-LIF | 0.51 | 0.8 | 0.7 | 0.78 | 0.51 | 0.62 |
| 13 | $\frac{H}{8} \times \frac{W}{8}$ | 512 | QConv-BN-LIF | 0.51 | 0.8 | 0.7 | 0.78 | 0.51 | 0.62 |
| 14 | - | - | MaxPool | - | | | | | |
| 15 | $\frac{H}{16} \times \frac{W}{16}$ | 512 | QConv-BN-LIF | 0.51 | 0.8 | 0.7 | 0.78 | 0.51 | 0.62 |
| 16 | $\frac{H}{16} \times \frac{W}{16}$ | 512 | QConv-BN-LIF | 0.51 | 0.8 | 0.7 | 0.78 | 0.51 | 0.62 |
| 17 | $\frac{H}{16} \times \frac{W}{16}$ | 512 | QConv-BN-LIF | - | - | - | - | - | - |

Table 10: Detailed network architecture and the channel pruning ratio for ResNet20.

| Layer | Resolution | Channel | Module | Channel Pruning Ratio CIFAR-10 / 100 | |
|---|---|---|---|---|---|
| | | | | 6.22M / 6.27M | 3.87M / 3.92M |
| conv0 | $H \times W$ | 64 | Conv-BN-LIF | 0.1 | 0.1 |
| Layer1.0 | $H \times W$ | 128 | QConv-BN-LIF | 0.3 | 0.35 |
| | | | QConv-BN-LIF | 0.3 | 0.35 |
| Layer1.1 | $H \times W$ | 128 | QConv-BN-LIF | 0.6 | 0.75 |
| | | | QConv-BN-LIF | 0.3 | 0.35 |
| Layer1.2 | $H \times W$ | 128 | QConv-BN-LIF | 0.6 | 0.75 |
| | | | QConv-BN-LIF | 0.3 | 0.35 |
| Layer2.0 | $\frac{H}{2} \times \frac{W}{2}$ | 256 | QConv-BN-LIF | 0.6 | 0.75 |
| | | | QConv-BN-LIF | 0.6 | 0.75 |
| Layer2.1 | $\frac{H}{2} \times \frac{W}{2}$ | 256 | QConv-BN-LIF | 0.6 | 0.75 |
| | | | QConv-BN-LIF | 0.6 | 0.75 |
| Layer2.2 | $\frac{H}{2} \times \frac{W}{2}$ | 256 | QConv-BN-LIF | 0.6 | 0.75 |
| | | | QConv-BN-LIF | 0.6 | 0.75 |
| Layer3.0 | $\frac{H}{4} \times \frac{W}{4}$ | 512 | QConv-BN-LIF | 0.6 | 0.75 |
| | | | QConv-BN-LIF | - | - |
| Layer3.1 | $\frac{H}{4} \times \frac{W}{4}$ | 512 | QConv-BN-LIF | 0.6 | 0.75 |
| | | | QConv-BN-LIF | - | - |
| Layer3.2 | $\frac{H}{4} \times \frac{W}{4}$ | 512 | QConv-BN-LIF | 0.6 | 0.75 |
| | | | QConv-BN-LIF | - | - |

Table 11: Detailed network architecture and the channel pruning ratio for VGGSNN.

| Layer | Resolution | Channel | Module | Channel Pruning Ratio DVS-CIFAR10 | | |
|---|---|---|---|---|---|---|
| | | | | 1.46M | 0.9M | 0.25M |
| 1 | $H \times W$ | 64 | Conv-BN-LIF | 0.5 | 0.5 | 0.82 |
| 2 | $H \times W$ | 128 | QConv-BN-LIF | 0.5 | 0.5 | 0.82 |
| 3 | - | | | MaxPool | - | |
| 4 | $\frac{H}{2} \times \frac{W}{2}$ | 256 | QConv-BN-LIF | 0.5 | 0.5 | 0.82 |
| 5 | $\frac{H}{2} \times \frac{W}{2}$ | 256 | QConv-BN-LIF | 0.7 | 0.8 | 0.93 |
| 6 | - | | | MaxPool | - | |
| 7 | $\frac{H}{4} \times \frac{W}{4}$ | 512 | QConv-BN-LIF | 0.7 | 0.8 | 0.93 |
| 8 | $\frac{H}{4} \times \frac{W}{4}$ | 512 | QConv-BN-LIF | 0.7 | 0.8 | 0.93 |
| 9 | - | | | MaxPool | - | |
| 10 | $\frac{H}{8} \times \frac{W}{8}$ | 512 | QConv-BN-LIF | 0.7 | 0.8 | 0.93 |
| 11 | $\frac{H}{8} \times \frac{W}{8}$ | 512 | QConv-BN-LIF | - | - | - |

**Model size calculation**   The model size is computed by aggregating the storage requirements of both quantized and full precision parameters, as expressed by the following equation (Qin et al. (2022); Zhang et al. (2022)),

$$M = \text{Params} \times \text{Bitwidth} = \sum P_q \times B_q + \sum P_{fp} \times B_{fp}, \qquad (16)$$

where $P_q$ and $P_{fp}$ denote the quantized parameters and full precision parameters, respectively, while $B_q$ and $B_{fp}$ represent their corresponding bit widths. It is important to note that, in our experiments, full-precision weights are employed in both the initial convolutional layer and the final fully connected layer to ensure optimal performance (Zhang et al. (2021b); Ding et al. (2022)). We also take this configuration into account when calculating our model size.

