# OpenReview forum: "QP-SNN: Quantized and Pruned Spiking Neural Networks"
_ICLR.cc/2025/Conference — ICLR 2025 Poster_

### Official Review · Reviewer_AehT · 2024-10-31

**Soundness:** 3
**Presentation:** 3
**Contribution:** 3
**Rating:** 8
**Confidence:** 5

**Summary:**

This paper presents QP-SNN, a novel approach to creating efficient and hardware-friendly Spiking Neural Networks (SNNs) by combining uniform quantization and structured pruning. The authors first develop a baseline model, then address its performance limitations through two key innovations: (1) a weight rescaling (ReScaW) strategy for more effective bit-width utilization in quantization, and (2) a novel pruning criterion based on the singular value of spatiotemporal spike activities (SVS). The experimental results demonstrate that QP-SNN achieves state-of-the-art performance while significantly reducing model size.
Strengths

**Strengths:**

1. Theoretical Foundation:
- The analysis of bit-width utilization provides clear theoretical justification for the ReScaW strategy.
- The proposed SVS criterion is well-grounded in mathematical principles.

2. Method Innovation:
- The paper presents two novel techniques (ReScaW and SVS) that effectively address specific limitations in existing quantization and pruning approaches.
- The solutions are well-motivated by careful analysis of the underlying problems.


3. Experimental Validation:
- Comprehensive experiments across multiple datasets and architectures.
- Detailed ablation studies that validate each component's contribution.

**Weaknesses:**

1.The selection process for pruning rates appears arbitrary and lacks systematic justification.
2.This paper is only verified on the simple computer vision classification task.

**Questions:**

1. How does the computational cost of calculating singular values in the SVS criterion compare to existing pruning criteria? Is there a significant overhead during training?

2.What is the impact of the proposed methods on the network's spike sparsity?

3.How sensitive is the method to the choice of ε in the SVS-based pruning criterion?

How does the approach perform when applied to other types of neural networks beyond classification tasks, such as object detection?

---

> ### Author Response · Authors · 2024-11-23
> **Response to W1**
>
> Thank you very much for your recognition of our work and your insightful review. We will address each of your questions.
>
> > *W1: The selection process for pruning rates appears arbitrary and lacks systematic justification.*
>
> *Response to W1*: We acknowledge that the setting of pruning rates is arbitrary, but no tedious manual design is required. The suitable pruning rates of each layer can be determined through the orthogonal NAS-based approach. However, we decide not to do this, **aiming to maintain the simplicity and practicality of our method.** More importantly, **without requiring tedious manual design and parameter search, QP-SNN achieves state-of-the-art performance compared to advanced methods. This further demonstrates the effectiveness of our method.** Of course,  if the automated search mechanism is introduced into our method, the performance can be further improved. This will be an important focus of our future work.

---

> ### Author Response · Authors · 2024-11-23
> **Response to W2 &Q4**
>
> > *W2: This paper is only verified on the simple computer vision classification task.*
> >
> > *Q4: How does the approach perform when applied to other types of neural networks beyond classification tasks, such as object detection?*
>
> *Response to W2 &Q4*: Thank you for pointing out this issue. Our method can be applied to more complex vision tasks like object detection. The reason we choose the simple classification tasks is to facilitate a comprehensive comparison with advanced compression methods in SNNs [1,2]. To address your concern more effectively, **we have conducted additional experiments for the object detection task**. We select two remote the sensing dataset SSDD [3]. The SSDD dataset specifically focuses on ship detection imagery acquired through synthetic aperture radar. In our experiments, we adopt the YOLO-v3 detection architecture with ResNet10 as the backbone. During training, we perform the pruning operation on the backbone and employ the SGD optimizer with a polynomial decay learning rate schedule, initializing the learning rate at 1e-2 and training for 300 epochs. **Results are shown in the table below, which fully validates the effectiveness of QP-SNN for complex vision tasks.**
>
> | Dataset | Method         | Bit Width | Model size (MB) | mAP@0.5 |
> | :------ | -------------- | --------- | --------------- | ------- |
> | SSDD    | Full-precision | 32        | 19.29           | 96.8%   |
> | SSDD    | QP-SNN         | 4         | 2.15            | 97.10%  |
>
> >*[1] Shi, Xinyu, et al. Towards energy efficient spiking neural networks: An unstructured pruning framework. ICLR 2024.*
>
> >*[2] Li, Yaxin, et al. Towards efficient deep spiking neural networks construction with spiking activity based pruning. ICML 2024.*
>
> >*[3] Cheng, Gong, Junwei Han, and Xiaoqiang Lu. Remote sensing image scene classification: Benchmark and state of the art. Proceedings of the IEEE 105.10 (2017): 1865-1883.*

---

> ### Author Response · Authors · 2024-11-23
> **Response to Q1**
>
> > *Q1: How does the computational cost of calculating singular values in the SVS criterion compare to existing pruning criteria? Is there a significant overhead during training?*
>
> *Response to Q1*: Currently, the advanced methods in SNNs [1,2] perform pruning during the training process. In contrast, the proposed SVS criterion is applied only once before pruning, which does not involve the training process. More specifically, we first obtain a well-trained quantized model. Then, we use the SVS-based pruning criterion to assess the importance of each channel and prune redundant kernels based on the pruning rates. Finally, we fine-tune the pruned model. The SVD computation is not required during the fine-tuning stage, so the training efficiency is not affected.
>
> >*[1] Shi, Xinyu, et al. Towards energy efficient spiking neural networks: An unstructured pruning framework. ICLR 2024.*
>
> >*[2] Li, Yaxin, et al. Towards efficient deep spiking neural networks construction with spiking activity based pruning. ICML 2024.*

---

> ### Author Response · Authors · 2024-11-23
> **Response to Q2**
>
> > *Q2: What is the impact of the proposed methods on the network's spike sparsity?*
>
> *Response to Q2*: To analyze the impact of our method on the spike sparsity of the neural network, we compare the firing rates of the original SNN and QP-SNN across each layer. The comparison is performed on the CIFAR-10 dataset using the VGG-16 architecture and the CIFAR-100 dataset using the ResNet20 architecture, and the results are summarized in the table below.
>
> | Method         | Architecture | Avg fr | conv1  | layer1.0.lif1 | layer1.0.lif2 | layer1.1.lif1 | layer1.1.lif2 | layer1.2.lif1 | layer1.2.lif2 | layer2.0.lif1 | layer2.0.lif2 | layer2.1.lif1 |
> | -------------- | ------------ | ------ | ------ | ------------- | ------------- | ------------- | ------------- | ------------- | ------------- | ------------- | ------------- | ------------- |
> | Full precision | ResNet20     | 0.1926 | 0.3597 | 0.1993        | 0.2541        | 0.1488        | 0.2669        | 0.1313        | 0.2795        | 0.1973        | 0.2249        | 0.1109        |
> | QP-SNN         | ResNet20     | 0.2141 | 0.2792 | 0.2097        | 0.2525        | 0.2479        | 0.2523        | 0.2412        | 0.2469        | 0.3321        | 0.2115        | 0.1359        |
>
> | Method         | Architecture | layer2.1.lif2 | layer2.2.lif1 | layer2.2.lif2 | layer3.0.lif1 | layer3.0.lif2 | layer3.1.lif1 | layer3.1.lif2 | layer3.2.lif1 | layer3.2.lif2 |
> | -------------- | ------------ | ------------- | ------------- | ------------- | ------------- | ------------- | ------------- | ------------- | ------------- | ------------- |
> | Full precision | ResNet20     | 0.2172        | 0.1179        | 0.2110        | 0.1397        | 0.1420        | 0.1000        | 0.1207        | 0.0885        | 0.3498        |
> | QP-SNN         | ResNet20     | 0.2197        | 0.1224        | 0.2142        | 0.2264        | 0.1126        | 0.1590        | 0.1105        | 0.1987        | 0.2875        |
>
> | Method         | Architecture | Avg fr | layer1 | layer2 | layer3 | layer4 | layer5 | layer6 | layer7 | layer8 | layer9 | layer10 | layer11 | layer12 | layer13 |
> | -------------- | ------------ | ------ | ------ | ------ | ------ | ------ | ------ | ------ | ------ | ------ | ------ | ------- | ------- | ------- | ------- |
> | Full precision | VGG-16       | 0.1579 | 0.2939 | 0.0927 | 0.1895 | 0.0959 | 0.1443 | 0.1319 | 0.0761 | 0.1148 | 0.1426 | 0.1791  | 0.1993  | 0.1936  | 0.1993  |
> | QP-SNN         | VGG-16       | 0.1923 | 0.3152 | 0.2486 | 0.2436 | 0.1672 | 0.2092 | 0.1689 | 0.0778 | 0.1385 | 0.1428 | 0.1472  | 0.1916  | 0.1649  | 0.2845  |
>
> From the comparison, we observe that the firing rate of QP-SNN is marginally higher than that of the original model. Despite this slight increase, **our method significantly reduces storage and computational costs, since it achieves extreme model compression by unifying both pruning and quantization.** For example, on the CIFAR-10 dataset, the original model size is reduced by 98.74%, SOPs are decreased by 78.69%, and power consumption is lowered by 77.45%.
>
> | Dataset   | Architecture | Connection | Bit Width | Model size | SOPs(M) | Power(mJ) |
> | --------- | ------------ | ---------- | --------- | ---------- | ------- | --------- |
> | CIFAR-10  | VGG-16       | 100%       | 32        | 58.88      | 54.6    | 0.204     |
> |           | VGG-16       | 9.61%      | 4         | 0.74       | 11.63   | 0.046     |
> | CIFAR-100 | ResNet20     | 100%       | 32        | 68.4       | 415.64  | 0.756     |
> |           | ResNet20     | 22.92%     | 4         | 2.17       | 131.53  | 0.126     |

---

> ### Author Response · Authors · 2024-11-23
> **Response to Q3**
>
> >*Q3: How sensitive is the method to the choice of ε in the SVS-based pruning criterion?*
>
> *Response to Q3*: ε is the threshold used to distinguish significant singular values, and its selection should be based on the specific task demands. A higher ε value zeros out small singular values, which helps reduce noise and prevent overfitting, making it suitable for scenarios with high data noise. A lower ε value preserves more fine-grained features, which helps prevent information loss, making it ideal for tasks that demand high accuracy. In summary, setting ε too high may overlook subtle linear relationships in the matrix, while setting it too low may introduce redundant information like noise. In this work, we set ε to a relatively small value to retain as much useful information as possible.

---

> > ### Comment · Reviewer_AehT · 2024-11-24
> > **one more question**
> >
> > I appreciate your response and extra experiments. Most of the concerns have been addressed, but I still have one question.
> >
> > The authors have added experiments on the object detection task, but the selected SSDD dataset is relatively simple, focusing primarily on ship detection for a specific scenario. It is recommended that the authors consider validation on more challenging datasets, such as those containing multiple categories and complex scenes.

---

> ### Author Response · Authors · 2024-11-25
> **Response**
>
> >*Q: The authors have added experiments on the object detection task, but the selected SSDD dataset is relatively simple, focusing primarily on ship detection for a specific scenario. It is recommended that the authors consider validation on more challenging datasets, such as those containing multiple categories and complex scenes.*
>
>
>
> Thank you for your insightful feedback and the time you dedicated to evaluating our work. To better answer your question, we have conducted additional experiments on the NWPU VHR-10 dataset[1], which contains 10 object categories (airplane, ship, storage tank, baseball diamond, tennis court, basketball court, ground track field, harbor, bridge and vehicle) in complex scenes with various backgrounds.
>
> **Experimental details**: We adopt the YOLO-v3 detection architecture with ResNet10 as the backbone. During training, we perform the pruning operation on the backbone and employ the SGD optimizer with a polynomial decay learning rate schedule, initializing the learning rate at 1e-2 and training for 300 epochs. All our experiments are conducted on four NVIDIA A100 GPUs, and the code has been included in the supplementary materials for reproducibility.
>
> **Experimental results**: The results are summarized in the table below. Notably, on this more complex object detection dataset, QP-SNN still achieves a significant reduction in model size while maintaining satisfactory detection performance. This fully demonstrates the potential of our approach to extend to more challenging tasks. We have added the results and visualizations on this dataset in the revised manuscript.
>
> | Dataset     | Method         | Bit width | Model size | mAP@0.5 |
> | ----------- | -------------- | --------- | ---------- | ------- |
> | NWPU VHR-10 | Full-precision | 32        | 19.29      | 89.89%  |
> | NWPU VHR-10 | QP-SNN         | 4         | 2.15       | 86.68%  |
>
> > *[1] Cheng, Gong, Junwei Han, and Xiaoqiang Lu. "Remote sensing image scene classification: Benchmark and state of the art." Proceedings of the IEEE 105.10 (2017): 1865-1883.*

---

> > ### Comment · Reviewer_AehT · 2024-11-26
> > **thanks**
> >
> > thanks for response. my concern has been well addressed. I have raised my score.

---

> > > ### Author Response · Authors · 2024-11-26
> > >
> > > Thank you very much for your recognition. We are pleased to see that your concerns have been effectively addressed. Once again, we greatly appreciate your valuable review and feedback, which have been immensely helpful to us.

---

### Official Review · Reviewer_Likb · 2024-11-01

**Soundness:** 2
**Presentation:** 3
**Contribution:** 1
**Rating:** 5
**Confidence:** 4

**Summary:**

The paper proposes two technologies including a weight-rescaling strategy and an SVS-based pruning criterion for weight quantization and structured pruning of SNN respectively. These technologies reduce the model size significantly while maintains the SNN accuracy.

**Strengths:**

1. The paper develops a hardware-efficient and lightweight QP-SNN baseline by integrating uniform quantization and structured pruning.
2. The weight-rescaling strategy and the SVS-based pruning criterion works well on SNN benchmarks.

**Weaknesses:**

1. The proposed technologies including the weight-rescaling strategy and the SVS-based pruning criterion do not fully utilize the unique characteristics of SNN, such as the sparsity to reduce synaptic operation and temporal information to maintain the SNN accuracy. The technologies can also be used in ANN models.
2. The comparison with SNNs that are not compressed is missing, such as SEW-ResNet[1] and MS-ResNet[2]. Compared to these works, the accuracy degradation is still large.
3. The results that the saving of synaptic operation is missing.

[1] Fang W, Yu Z, Chen Y, et al. Deep residual learning in spiking neural networks[J]. Advances in Neural Information Processing Systems, 2021, 34: 21056-21069.
[2] Hu Y, Deng L, Wu Y, et al. Advancing spiking neural networks toward deep residual learning[J]. IEEE Transactions on Neural Networks and Learning Systems, 2024.

**Questions:**

1. What is the effectiveness of this work in saving synaptic operation in SNN inference? The author can use the code in https://github.com/zhouchenlin2096/Spikingformer/tree/master/energy_consumption_calculation to provide the detailed number of synaptic operations of the compressed SNN in this work.

---

> ### Author Response · Authors · 2024-11-23
> **Response to W1**
>
> We sincerely appreciate the time you have dedicated to reviewing our manuscript and for providing such constructive feedback. We will address each of your concerns.
>
> > *W1: The proposed technologies including the weight-rescaling strategy and the SVS-based pruning criterion do not fully utilize the unique characteristics of SNN. The technologies can also be used in ANN models.*
>
> *Response to W1*: As you pointed out, both proposed methods can also be used in ANNs. **However, the proposed SVS pruning criterion is more suitable for SNN.** Specifically, in SNNs, the SVS-based pruning criterion employs SVD on the spatio-temporal spike matrix, where the matrix values are constrained to a few discrete levels (for example, when \(T=4\), the matrix values can only be \{0, 1/4, 1/2, 3/4, 1\}). Therefore, the advantage of SVS -based pruning in SNNs can be summarized as two aspects.
>
> - **First, SVS-based pruning criterion in SNNs offers significant computational efficiency advantages when considering the hardware deployment.** SVD computations for continuous matrices typically require high-precision numerical processing, as matrix values are real numbers and can vary widely. In contrast, discrete matrices use a limited number of values, allowing for more compact data representation and reduced memory usage. This significantly improves computational efficiency during large-scale matrix decomposition.
>
> - **Second, SVS-based pruning criterion in SNNs demonstrates more stable kernel importance evaluation on discrete matrices.** The SVD decomposition performed on continuous matrices is more sensitive to noise perturbations due to the high precision representation, which results in fluctuations in channel importance scores. In contrast, discrete matrices can more effectively suppress small noise changes due to their lower precision, leading to more stable importance evaluation.

---

> ### Author Response · Authors · 2024-11-23
> **Response to W2**
>
> > *W2: The comparison with SNNs that are not compressed is missing. Compared to these works, the accuracy degradation is still large.*
>
> *Response to W2*: Thank you for pointing out our issue. We have added a comprehensive comparison of QP-SNN with the corresponding uncompressed SNN.
> - We acknowledge that our method exhibits accuracy loss compared to uncompressed SNNs. **However,** **this performance degradation is a common challenge in the field of model compression**. Therefore, QP-SNN that compresses the network by reducing both parameter bit-width and structures will inevitably bring the performance drop. **Fortunately, QP-SNN demonstrates satisfactory performance under extreme compression ratios.** For example, on the CIFAR-10 dataset, under the extreme connection ratio of 9.61%, QP-SNN reduces the model size by 98.74%, SOPs by 78.69%, and power consumption by 77.45%, while the accuracy decreases by only 2.44%. **This trade-off between performance degradation and resource efficiency is highly advantageous in edge computing scenarios.**
>
> | Dataset   | Architecture | Connection | Bit Width | Model size | SOPs(M) | Power(mJ) | Accuracy(%) |
> | :-------- | ------------ | ---------- | --------- | ---------- | ------- | --------- | ----------- |
> | CIFAR-10  | VGG-16       | 100%       | 32        | 58.88      | 54.6    | 0.204     | 93.63%      |
> |           | VGG-16       | 9.61%      | 4         | 0.74       | 11.63   | 0.046     | 91.19%      |
> | CIFAR-100 | ResNet20     | 100%       | 32        | 68.4       | 415.64  | 0.756     | 79.49%      |
> |           | ResNet20     | 22.92%     | 4         | 2.17       | 131.53  | 0.126     | 74.73%      |
>
>
> Despite the accuracy loss compared with uncompressed SNN, **QP-SNN achieves state-of-the-art performance and efficiency among compressed SNNs**. This can be observed from the following table. Therefore, QP-SNN underscores its potential for enhancing SNN deployment in edge intelligence computing.
>
>
> | Method                      | Network  | Bit Width | Hardware friendly | Time step | Model size (MB) | Accuracy |
> | --------------------------- | -------- | --------- | ----------------- | --------- | --------------- | -------- |
> | Chowdhury et al. (2021) [1] | VGG-9    | 5         | Yes               | 25        | 12.59           | 88.60%   |
> | Deng et al. (2021) [2]      | 7Conv2FC | 3         | No                | 8         | 5.84            | 87.59%   |
> | Shi et al. (2024) [3]       | 6Conv2FC | 32        | No                | 8         | 28.4            | 90.65%   |
> | Li et al. (2024) [4]        | VGG-16   | 32        | Yes               | 4         | 5.68            | 90.26%   |
> | QP-SNN                      | VGG-16   | 2         | Yes               | 4         | 1.10            | 91.61%   |
>
> >*[1] Chowdhury, Sayeed Shafayet, Isha Garg, and Kaushik Roy. Spatio-temporal pruning and quantization for low-latency spiking neural networks. IJCNN 2021.*
>
> >*[2] Deng, Lei, et al. Comprehensive snn compression using admm optimization and activity regularization. TNNLS 2021.*
>
> >*[3] Shi, Xinyu, et al. Towards energy efficient spiking neural networks: An unstructured pruning framework. ICLR 2024.*
>
> >*[4] Li, Yaxin, et al. Towards efficient deep spiking neural networks construction with spiking activity based pruning. ICML 2024.*

---

> ### Author Response · Authors · 2024-11-23
> **Response to W3 and Q1**
>
> *W3: The results that the saving of synaptic operation is missing.*
>
> *Q1: What is the effectiveness of this work in saving synaptic operation in SNN inference?*
>
> *Response to W3 and Q1*: Thank you for your valuable suggestions and guidance. We have added SOP comparisons of QP-SNN with full-precision uncompressed SNNs and related compressed studies.
>
> - We first provide the comparison results between our method and full-precision uncompressed counterparts. As shown in the table below, QP-SNN demonstrates satisfactory performance under extreme compression ratios. For example, on the CIFAR-10 dataset, under the extreme connection ratio of 9.61%, QP-SNN reduces the model size by 98.74%, SOPs by 78.69%, and power consumption by 77.45%, while the accuracy decreases by only 2.44%. **This trade-off between performance degradation and resource efficiency is highly advantageous in edge computing scenarios.**
>
> | Dataset   | Architecture | Connection | Bit Width | Model size | SOPs(M) | Power(mJ) | Accuracy(%) |
> | :-------- | ------------ | ---------- | --------- | ---------- | ------- | --------- | ----------- |
> | CIFAR-10  | VGG-16       | 100%       | 32        | 58.88      | 54.6    | 0.204     | 93.63%      |
> |           | VGG-16       | 9.61%      | 4         | 0.74       | 11.63   | 0.046     | 91.19%      |
> | CIFAR-100 | ResNet20     | 100%       | 32        | 68.4       | 415.64  | 0.756     | 79.49%      |
> |           | ResNet20     | 22.92%     | 4         | 2.17       | 131.53  | 0.126     | 74.73%      |
>
> - We then add the SOP comparison results between our method with related compressed studies on the CIFAR-10 dataset. Experimental results are shown in the following table. It can be seen that **QP-SNN exhibits competitive SOPs compared to compression work, while exhibiting extremely low model size due to quantization.** Moreover, it is worth noting that the advanced works [1,2] focus on unstructured pruning, which typically achieves higher sparsity and performance but requires specialized hardware support. In contrast, **our QP-SNN adopts uniform quantization and structured pruning, balancing the advantages of sparsity, performance, and hardware compatibility.**
>
> | Method         | Architecture | Time step | Hardware Friendly | Model size | SOPs(M) |
> | :------------- | ------------ | --------- | ----------------- | ---------- | ------- |
> | ADMM [2]       | 7 Conv, 2 FC | 8         | No                | 62.16      | 107.97  |
> | Shi et al. [1] | 6 Conv, 2 FC | 8         | No                | 33.76      | 11.98   |
> | Li et al. [3]  | VGG-16       | 4         | Yes               | 5.68       | -       |
> | QP-SNN         | VGG-16       | 4         | Yes               | 0.74       | 11.63   |
>
> >*[1] Shi, Xinyu, et al. Towards energy efficient spiking neural networks: An unstructured pruning framework. ICLR 2024.*
>
> >*[2] Deng, Lei, et al. Comprehensive snn compression using admm optimization and activity regularization. TNNLS 2021.*
>
> >*[3] Li, Yaxin, et al. Towards efficient deep spiking neural networks construction with spiking activity based pruning. ICML 2024.*

---

> ### Author Response · Authors · 2024-11-25
>
> Dear Reviewer Likb,
>
> We sincerely appreciate your time and effort in reviewing our manuscript and offering valuable suggestions. As the author-reviewer discussion phase is drawing to a close, we would like to confirm whether our responses have effectively addressed your concerns. We have provided detailed responses to your concerns, and we hope they have adequately addressed your issues. If you require further clarification or have any additional concerns, please do not hesitate to contact us. We are more than willing to continue our communication with you.
>
> Best regards.

---

> ### Comment · Reviewer_Likb · 2024-11-26
>
> Thanks for the authors' reply.  The supplementary experiments in rebuttal are still unconvinced. For example, why is the #SOP of VGG16 smaller than ResNet20? The input size of CIFAR10 and CIFAR100 is the same. The results are not correct. I will keep my score.

---

> > ### Author Response · Authors · 2024-11-26
> >
> > Thank you very much for your reply. We have included the code and logs for our supplementary experiments in the supplement to ensure reproducibility.
> >
> > Regarding the issue you raised, **although VGG-16 and ResNet20 have the same input shape, the intermediate feature maps of VGG-16 are much smaller than those of ResNet20, which results in lower #SOPs for VGG-16.** We present the shape of the feature maps for each layer of VGG-16 and ResNet20 in the following table. As can be clearly seen, the feature map sizes in VGG-16 are significantly smaller. The number of the synaptic operations is calculated as [1,2],
> > $$
> > SOP^l=fr\times T\times MACs,\\
> > {MACs} = H_{\text{out}} \times W_{\text{out}} \times C_{\text{in}} \times K_h \times K_w \times C_{\text{out}},
> > $$
> > where $H_{\text{out}}, W_{\text{out}}$ are the height and width of the output feature map, $C_{\text{in}}$ is the number of input channels, $K_h, K_w$ are the height and width of the convolution kernel, and $C_{\text{out}}$ is the number of output channels. It can be observed that the MAC calculation is closely related to the size of the output feature map; the larger the output feature map, the greater the total number of MACs.
> >
> > > *[1] Zhou, Chenlin, et al. "Spikingformer: Spike-driven residual learning for transformer-based spiking neural network." arXiv preprint arXiv:2304.11954 (2023).*
> > >
> > > *[2] Zhou, Chenlin, et al. QKFormer: Hierarchical Spiking Transformer using QK Attention. NeurIPS 2024.*
> >
> > **ResNet20**
> >
> > | Layer                         | Output Shape (Conv: [B, C, H, W]; Linear: [B, C, L]) |
> > | ----------------------------- | ---------------------------------------------------- |
> > | conv1_s.layer.module          | [1, 64, 32, 32]                                      |
> > | layer1.0.conv1_s.layer.module | [1, 128, 32, 32]                                     |
> > | layer1.0.conv2_s.layer.module | [1, 128, 32, 32]                                     |
> > | layer1.1.conv1_s.layer.module | [1, 128, 32, 32]                                     |
> > | layer1.1.conv2_s.layer.module | [1, 128, 32, 32]                                     |
> > | layer1.2.conv1_s.layer.module | [1, 128, 32, 32]                                     |
> > | layer1.2.conv2_s.layer.module | [1, 128, 32, 32]                                     |
> > | layer2.0.conv1_s.layer.module | [1, 256, 16, 16]                                     |
> > | layer2.0.conv2_s.layer.module | [1, 256, 16, 16]                                     |
> > | layer2.1.conv1_s.layer.module | [1, 256, 16, 16]                                     |
> > | layer2.1.conv2_s.layer.module | [1, 256, 16, 16]                                     |
> > | layer2.2.conv1_s.layer.module | [1, 256, 16, 16]                                     |
> > | layer2.2.conv2_s.layer.module | [1, 256, 16, 16]                                     |
> > | layer3.0.conv1_s.layer.module | [1, 512, 8, 8 ]                                      |
> > | layer3.0.conv2_s.layer.module | [1, 512, 8, 8 ]                                      |
> > | layer3.1.conv1_s.layer.module | [1, 512, 8, 8 ]                                      |
> > | layer3.1.conv2_s.layer.module | [1, 512, 8, 8 ]                                      |
> > | layer3.2.conv1_s.layer.module | [1, 512, 8, 8 ]                                      |
> > | layer3.2.conv2_s.layer.module | [1, 512, 8, 8 ]                                      |
> > | fc.module                     | [1, 100]                                             |
> >
> > **VGG-16**
> >
> > | Layer                 | Output Shape (Conv: [B, C, H, W]; Linear: [B, C, L]) |
> > | --------------------- | ---------------------------------------------------- |
> > | convbn0.layer.module  | [1, 64, 32, 32]                                      |
> > | convbn1.layer.module  | [1, 64, 32, 32]                                      |
> > | convbn3.layer.module  | [1, 128, 16, 16]                                     |
> > | convbn4.layer.module  | [1, 128, 16, 16]                                     |
> > | convbn6.layer.module  | [1, 256, 8, 8]                                       |
> > | convbn7.layer.module  | [1, 256, 8, 8]                                       |
> > | convbn8.layer.module  | [1, 256, 8, 8]                                       |
> > | convbn10.layer.module | [1, 512, 4, 4]                                       |
> > | convbn11.layer.module | [1, 512, 4, 4]                                       |
> > | convbn12.layer.module | [1, 512, 4, 4]                                       |
> > | convbn14.layer.module | [1, 512, 2, 2]                                       |
> > | convbn15.layer.module | [1, 512, 2, 2]                                       |
> > | convbn16.layer.module | [1, 512, 2, 2]                                       |
> > | linear1.module        | [1, 10]                                              |

---

> > > ### Comment · Reviewer_Likb · 2024-11-27
> > >
> > > Thanks for the author's reply. I will raise my score.

---

> > > > ### Author Response · Authors · 2024-11-28
> > > >
> > > > Dear Reviewer Likb,
> > > >
> > > > Thanks very much for your reply and recognition. We are happy to see that your concerns have been addressed. We will carefully revise the manuscript based on your comments and those of all other Reviewers to make it better.
> > > >
> > > > Best regards.

---

### Official Review · Reviewer_LMLx · 2024-11-02

**Soundness:** 2
**Presentation:** 3
**Contribution:** 2
**Rating:** 6
**Confidence:** 4

**Summary:**

This paper presents QP-SNN, which integrates quantization and structure pruning methods. The quantization approach is based on hardware-friendly uniform quantization, while the pruning method employs SVD techniques. QP-SNN has been validated on classification datasets, achieving state-of-the-art performance.

**Strengths:**

1. The writing of this paper is good and easy to read.

2. The experiments in the paper demonstrate the effectiveness of the proposed ReScaW algorithm and SVD-based pruning.

**Weaknesses:**

1. The ReScaW-based uniform quantization lacks innovation, as it is merely a combination of uniform quantization and weight scaling [1].

2. The SVD-based pruning appears to be an application of SVD pruning in SNNs. There are many methods utilizing SVD decomposition or selecting important singular values for pruning [2][3]. Please clarify the specific advantages of applying this in SNNs.

3. The SVD decomposition method introduces additional time complexity. How much does it specifically affect the training time?

4. The experiments are insufficient from several perspectives. Firstly, why use the 2021 SEW-ResNet and Spiking ResNet instead of the latest SNN models? Secondly, this method has only been tested on simple classification tasks; without testing on more complex downstream tasks. Thus, it is difficult to prove the potential applicability of this work.

5. As an efficiency-focused study, why does the paper not present sufficiently robust efficiency metrics, such as OPs/SOPs, training time, inference speed, and energy consumption?

6. Why there is no presentation of pruning ratio statistics in the main text, which obscures the specific effectiveness of this aspect?

**Reference**

[1] AWQ: ACTIVATION-AWARE WEIGHT QUANTIZATION FOR ON-DEVICE LLM COMPRESSION AND ACCELERATION.

[2] Learning Low-rank Deep Neural Networks via Singular Vector Orthogonality Regularization and Singular Value Sparsification.

[3] HRank: Filter Pruning using High-Rank Feature Map.

**Questions:**

Please see the weaknesses.

---

> ### Author Response · Authors · 2024-11-23
> **Response to W1**
>
> We are deeply grateful for your time in reviewing our manuscript. Your insights will be of great help to us in improving the quality of our paper.
>
> > *W1: The ReScaW-based uniform quantization lacks innovation, as it is merely a combination of uniform quantization and weight scaling[1].*
> >
> > [1] Lin, Ji, et al. "AWQ: Activation-aware Weight Quantization for On-Device LLM Compression and Acceleration." *Proceedings of Machine Learning and Systems* 6 (2024): 87-100.
>
> *Response to W1*: Yes, the ReScaW-based method is a combination of uniform quantization and weight scaling. However, we want to make a statement about its innovativeness and effectiveness. *First is the innovation of the ReScaW-based method.* Regarding the AWQ[1] you mentioned, although both AWQ and our work involve quantization and introduce scaling factors, there are significant differences between them, which can be summarized in two aspects.
>
> - **On the one hand, the problems they aim to solve are different.** AWQ aims to identify and protect a small fraction (0.1%–1%) of salient weights, thereby reducing the quantization error of LLMs. In contrast, ReScaW focuses on improving the bit-width usage efficiency of vanilla uniform quantization, thereby addressing the reduced discrimination of quantized weights. The former emphasizes reducing quantization errors for a small fraction of salient weights, while the latter is dedicated to optimizing overall bit-width usage efficiency.
>
> - **On the other hand, their technical approaches differ markedly.** AWQ employs an automatic search method during the training process to determine the scaling factor, with the search space defined based on the activation magnitude. In contrast, the ReScaW strategy utilizes weight distribution characteristics before quantization, such as maximum values, percentiles, and mean of the L1 norm. The scaling coefficient in our ReScaW does not involve optimization, automatic search, or activation values.
>
> *Second is the effectiveness of the ReScaW-based method.* By combining uniform quantization with a scaling coefficient $\gamma$, it effectively addresses the performance degradation caused by inefficient bit-width usage in vanilla uniform quantization. To clearly illustrate the effectiveness of the ReScaW, we present the comparative results in the following table, where **incorporating ReScaW into the baseline improves performance by 4.24%.** More importantly, this scaling coefficient can be directly fused into the convolutional layer during inference. This simple method **avoids the additional computation and storage overhead of complex strategies,** which offers an efficient solution for resource-constrained edge deployments.
>
> | Model                 | Accuracy                |
> | --------------------- | ----------------------- |
> | baseline              | 69.16%                  |
> | baseline r.w./ ReScaW | 73.40% (baseline+4.24%) |

---

> ### Author Response · Authors · 2024-11-23
> **Response to W2**
>
> > *W2: The SVD-based pruning appears to be an application of SVD pruning in SNNs. There are many methods utilizing SVD decomposition or selecting important singular values for pruning. Please clarify the specific advantages of applying this in SNNs.*
>
> *Response to W2*:  Yes, the SVD-based pruning method can also be applied to ANNs, but it has unique advantages in SNNs. More specifically, in SNNs, the SVS-based pruning criterion employs SVD on the spatio-temporal spike matrix, where the matrix values are constrained to a few discrete levels (for example, when \(T=4\), the matrix values can only be \{0, 1/4, 1/2, 3/4, 1\}). If this pruning method is used in ANNs, it should be applied to full-precision feature maps accordingly. Therefore, the advantage of SVD-based pruning in SNNs can be summarized as two aspects.
>
> - **First, SVD-based pruning in SNNs offers significant computational efficiency advantages when considering the hardware deployment.** SVD computations for continuous matrices typically require high-precision numerical processing, as matrix values are real numbers and can vary widely. In contrast, discrete matrices use a limited number of values, allowing for more compact data representation and reduced memory usage. This significantly improves computational efficiency during large-scale matrix decomposition.
> - **Second, SVD-based pruning in SNNs demonstrates more stable kernel importance evaluation on discrete matrices**. The SVD decomposition performed on continuous matrices is more sensitive to noise perturbations due to the high precision representation, which results in fluctuations in channel importance scores. In contrast, discrete matrices can more effectively suppress small noise changes due to their lower precision, leading to more stable importance evaluation.

---

> ### Author Response · Authors · 2024-11-23
> **Response to W3**
>
> > *W3: The SVD decomposition method introduces additional time complexity. How much does it specifically affect the training time?*
>
> *Response to W3*: **SVD on the spatiotemporal spike activity is applied only once, it does not impact the training process.** Specifically, we first obtain a well-trained quantized model. Then, we use the SVS-based pruning criterion to evaluate the importance of each channel and prune redundant kernels based on the pruning rate. Finally, we fine-tune the pruned model; during this fine-tuning phase, SVD is not used again.

---

> ### Author Response · Authors · 2024-11-23
> **Response to W4**
>
> > *W4: The experiments are insufficient from several perspectives. Firstly, why use the 2021 SEW-ResNet and Spiking ResNet instead of the latest SNN models? Secondly, this method has only been tested on simple classification tasks; without testing on more complex downstream tasks. Thus, it is difficult to prove the potential applicability of this work.*
>
> *Response to W4*: Thank you for pointing out this issue. Our method can be extended to more complex architectures like the Transformer and applied to tasks such as object detection. The reason we choose the ResNet architecture and simple classification tasks is to facilitate a comprehensive comparison with advanced compression methods in SNNs [1,2]. To address your concern more effectively, **we have conducted two additional experiments: (1) using the Spikingformer [3] architecture on the CIFAR-10 dataset, and (2) applying our method to an object detection task.** The results of these two experiments are presented in the following.
>
> - We first select the Spikingformer-4-384 structure [3] and validate it on the CIFAR-100 dataset. As shown in the table below, **our method achieves a 87.93% reduction in model size, a 55.48% decrease in SOPs, and a 55.64% reduction in energy consumption, while maintaining performance of 76.94%**. These results fully validate the effectiveness of QP-SNN for complex Spiking Transformer structures.
>
> | Architecture        | Method         | Connection | Bit Width | Model size(MB) | SOPs(M) | Power(mJ) | Accuracy |
> | :------------------ | -------------- | ---------- | --------- | -------------- | ------- | --------- | -------- |
> | Spikingformer-4-384 | Full-precision | 100%       | 16        | 18.64          | 292.14  | 0.266     | 79.09%   |
> | Spikingformer-4-384 | QP-SNN         | 44.74%     | 4         | 2.25           | 130.05  | 0.118     | 76.94%   |
>
> - We then conduct object detection experiments on the remote sensing dataset SSDD [4]. The SSDD dataset specifically focuses on ship detection imagery acquired through synthetic aperture radar. In our experiments, we adopt the YOLO-v3 detection architecture with ResNet10 as the backbone. During training, we perform the pruning operation on the backbone and employ the SGD optimizer with a polynomial decay learning rate schedule, initializing the learning rate at 1e-2 and training for 300 epochs. Results are shown in the table below, which demonstrate that **QP-SNN achieves a reduction in model size by 88.85%, while increasing mAP@0.5 by 0.3% compared to the full-precision uncompressed model.** This fully validates the effectiveness of QP-SNN for complex tasks.
>
> | Dataset | Method         | Bit Width | Model size (MB) | mAP@0.5 |
> | ------- | -------------- | --------- | --------------- | ------- |
> | SSDD    | Full-precision | 32        | 19.29           | 96.80%  |
> | SSDD    | QP-SNN         | 4         | 2.15            | 97.10%  |
>
>  **In summary, through these two experiments, we demonstrate that our method remains effective on more complex architectures and tasks.**
>
> > *[1] Shi, Xinyu, et al. Towards energy efficient spiking neural networks: An unstructured pruning framework. ICLR 2024.*
>
> > *[2] Li, Yaxin, et al. Towards efficient deep spiking neural networks construction with spiking activity based pruning. ICML 2024.*
>
> > *[3] Zhou, Chenlin, et al. Spikingformer: Spike-driven residual learning for transformer-based spiking neural network. arXiv preprint arXiv:2304.11954 (2023).*
>
> > *[4] Wang, Yuanyuan, et al. A SAR dataset of ship detection for deep learning under complex backgrounds. remote sensing 11.7 (2019): 765.*

---

> ### Author Response · Authors · 2024-11-23
> **Response to W5**
>
> >*W5: As an efficiency-focused study, why does the paper not present sufficiently robust efficiency metrics, such as OPs/SOPs, training time, inference speed, and energy consumption?*
>
> *Response to W5*: Thank you for pointing out this issue and for your valuable suggestions. Model compression aims to optimize efficiency during the inference phase, facilitating efficient deployment on resource-constrained devices. To this end, we have supplemented the key efficiency metrics of QP-SNN during inference, including model size, SOP, power consumption, the inference latency of the whole test dataset, and accuracy. Here, we present the results on the CIFAR-100 dataset with the ResNet20 structure. It can be observed that, **compared with the uncompressed full-precision model, our QP-SNN shows a significant efficiency advantage while maintaining competitive performance**.
>
> | Method         | Connection | Bit Width | Model size | SOPs(M) | Power(mJ) | Latency | Accuracy |
> | -------------- | ---------- | --------- | ---------- | ------- | --------- | ------- | -------- |
> | Full-precision | 100%       | 32        | 68.4       | 415.64  | 0.756     | 2.593s  | 79.49%   |
> | QP-SNN         | 22.92%     | 4         | 2.17       | 131.53  | 0.126     | 2.267s  | 74.73%   |

---

> ### Author Response · Authors · 2024-11-23
> **Response to W6**
>
> > *W6: Why there is no presentation of pruning ratio statistics in the main text, which obscures the specific effectiveness of this aspect?*
>
> *Response to W6*: We sincerely apologize for the inconvenience caused to you in understanding our paper. As per your proposal, we have included the channel pruning rates for various network architectures and datasets in Appendix D. In the following, we select the VGG-16 network on the CIFAR-10 dataset as a representative case, providing detailed channel pruning rates along with the corresponding experimental results.
>
> | Layer | Resolution  | Channel | Module           | Pruning ratio |
> | ----- | ----------- | ------- | ---------------- | --------------------- |
> | 1     | H × W       | 64      | Conv-BN-LIF      | 0.45                  |
> | 2     | H × W       | 64      | QuantConv-BN-LIF | 0.45                  |
> | 3     | -           | -       | MaxPool          | -                     |
> | 4     | H/2 × W/2   | 128     | QuantConv-BN-LIF | 0.45                  |
> | 5     | H/2 × W/2   | 128     | QuantConv-BN-LIF | 0.45                  |
> | 6     | -           | -       | MaxPool          | -                     |
> | 7     | H/4 × W/4   | 256     | QuantConv-BN-LIF | 0.45                  |
> | 8     | H/4 × W/4   | 256     | QuantConv-BN-LIF | 0.45                  |
> | 9     | H/4 × W/4   | 256     | QuantConv-BN-LIF | 0.45                  |
> | 10    | -           | -       | MaxPool          | -                     |
> | 11    | H/8 × W/8   | 512     | QuantConv-BN-LIF | 0.51                  |
> | 12    | H/8 × W/8   | 512     | QuantConv-BN-LIF | 0.51                  |
> | 13    | H/8 × W/8   | 512     | QuantConv-BN-LIF | 0.51                  |
> | 14    | -           | -       | MaxPool          | -                     |
> | 15    | H/16 × W/16 | 512     | QuantConv-BN-LIF | 0.51                  |
> | 16    | H/16 × W/16 | 512     | QuantConv-BN-LIF | 0.51                  |
> | 17    | H/16 × W/16 | 512     | QuantConv-BN-LIF | -                     |
>
> Under the above pruning rates, we present the performance of QP-SNN on the CIFAR-10 dataset and compare it with related studies. The experimental results are summarized in the table below. It can be observed that, **under such a low pruning rate combined with quantization, QP-SNN achieves state-of-the-art performance and efficiency, underscoring its potential for enhancing SNN deployment in edge intelligence computing**.
>
> | Method                      | Network  | Bit Width | Hardware friendly | Time step | Model size (MB) | Accuracy |
> | --------------------------- | -------- | --------- | ----------------- | --------- | --------------- | -------- |
> | Chowdhury et al. (2021) [1] | VGG-9    | 5         | Yes               | 25        | 12.59           | 88.60%   |
> | Deng et al. (2021) [2]      | 7Conv2FC | 3         | No                | 8         | 5.84            | 87.59%   |
> | Shi et al. (2024) [3]       | 6Conv2FC | 32        | No                | 8         | 28.4            | 90.65%   |
> | Li et al. (2024) [4]        | VGG-16   | 32        | Yes               | 4         | 5.68            | 90.26%   |
> | QP-SNN                      | VGG-16   | 2         | Yes               | 4         | 1.10            | 91.61%   |
>
> >*[1] Chowdhury, Sayeed Shafayet, Isha Garg, and Kaushik Roy. Spatio-temporal pruning and quantization for low-latency spiking neural networks. IJCNN 2021.*
>
> >*[2] Deng, Lei, et al. Comprehensive snn compression using admm optimization and activity regularization. TNNLS 2021.*
>
> >*[3] Shi, Xinyu, et al. Towards energy efficient spiking neural networks: An unstructured pruning framework. ICLR 2024.*
>
> >*[4] Li, Yaxin, et al. Towards efficient deep spiking neural networks construction with spiking activity based pruning. ICML 2024.*

---

> ### Author Response · Authors · 2024-11-25
> **Additional response to W4**
>
> >*W4: Secondly, this method has only been tested on simple classification tasks; without testing on more complex downstream tasks. Thus, it is difficult to prove the potential applicability of this work.*
>
> In addition to the SSDD dataset, we have also conducted additional experiments on the NWPU VHR-10 dataset[1], which contains 10 object categories (airplane, ship, storage tank, baseball diamond, tennis court, basketball court, ground track field, harbor, bridge and vehicle) in complex scenes with various backgrounds. We follow the same network architecture and experimental settings as conducted in the SSDD dataset. All our experiments are conducted on four NVIDIA A100 GPUs, and **the code has been included in the supplementary materials for reproducibility.**
>
> The results are summarized in the table below. Notably, on this more complex object detection dataset, QP-SNN still achieves a significant reduction in model size while maintaining satisfactory detection performance. **This fully demonstrates the potential of our approach to extend to more challenging tasks. We have added the results and visualizations on this dataset in the revised manuscript.**
>
> | Dataset     | Method         | Bit width | Model size | mAP@0.5 |
> | ----------- | -------------- | --------- | ---------- | ------- |
> | NWPU VHR-10 | Full-precision | 32        | 19.29      | 89.89%  |
> | NWPU VHR-10 | QP-SNN         | 4         | 2.15       | 86.68%  |
>
> > *[1] Cheng, Gong, Junwei Han, and Xiaoqiang Lu. "Remote sensing image scene classification: Benchmark and state of the art." Proceedings of the IEEE 105.10 (2017): 1865-1883.*

---

> ### Author Response · Authors · 2024-11-25
>
> Dear Reviewer LMLx,
>
> We sincerely appreciate your time and effort in reviewing our manuscript and offering valuable suggestions. As the author-reviewer discussion phase is drawing to a close, we would like to confirm whether our responses have effectively addressed your concerns. We have provided detailed responses to your concerns, and we hope they have adequately addressed your issues. If you require further clarification or have any additional concerns, please do not hesitate to contact us. We are more than willing to continue our communication with you.
>
> Best regards.

---

> > ### Comment · Reviewer_LMLx · 2024-11-27
> > **Official Comment by Reviewer LMLx**
> >
> > Dear authors,
> >
> > Thank you so much for your reply.
> >
> > My main concerns are regarding the novelty of the proposed methods (including ReScaW and SNN-based SVD) and the experimental shortcomings, particularly in the context of an efficiency-focused paper.
> >
> > *From the perspective of novelty:*
> >
> > - Firstly, I use AWQ as an example not to highlight how similar ReScaW and AWQ are in terms of their working principles, but rather to emphasize that rescaling is a very general technique, and it has been applied in numerous works. Therefore, I still believe the method lacks significant novelty, although its applicability to specific tasks is undoubtedly valuable.
> > - Secondly, I believe the explanation of the SVD method is reasonable, and the unique discrete representations in SNNs can indeed contribute to making the SVD process more stable.
> >
> > *From the perspective of work completeness:*
> >
> > As an efficiency-focused paper, it is a serious oversight to rely solely on classification as the primary experiment and to lack specific comparisons with efficiency metrics. However, I appreciate that the authors have supplemented the rebuttal with comparisons of metrics, experiments on the spiking transformer, and remote sensing experiments. I highly respect this effort, as adding additional experiments during the rebuttal phase is difficult and demanding.
> >
> > ---
> >
> > *I would like to offer the following suggestions for future work:*
> >
> > - Based on the current version, I encourage the authors to explore new aspects of novelty. While ReScaW and SNN-based SVD show good performance in the model, these are not new concepts but rather mature ones. The improvement achieved by successfully applying these mature concepts to specific tasks is not sufficient to demonstrate novelty.
> > - For efficiency-focused tasks, extensive and diverse experiments are necessary, as highlighted by other reviewers as well. I suggest that the authors consider introducing a wider range of tasks, including but not limited to detection, segmentation, and image generation. A good example is Fast-SNN [1]. This would provide stronger evidence of the effectiveness of the work.
> > - Efficiency metrics, including energy consumption, pruning rates, etc., need to be fully presented.
> >
> > In summary, although the paper has issues with novelty and experimental completeness, I believe some responses in the rebuttal are reasonable and sincere. I think the authors will improve the paper in future versions (e.g., supplement the additional experiments to the revised pdf). Therefore, I am willing to increase my score and am inclined to accept the paper.
> >
> > Best,
> >
> > Reviewer LMLx
> >
> > ---
> >
> > *Reference:*
> >
> > [1] Fast-SNN: Fast Spiking Neural Network by Converting Quantized ANN, TPAMI 2023.

---

> > > ### Author Response · Authors · 2024-11-28
> > >
> > > Dear Reviewer LMLx,
> > >
> > > We sincerely appreciate your recognition of our efforts. We are truly honored that our paper has encountered such a professional reviewer.
> > >
> > > *Main concerns:*
> > >
> > > - As per your suggestions, we will continue to explore and develop more innovative quantization and pruning methods that are better suited to SNNs, such as leveraging the sparsity and temporal dynamics inherent to SNNs. By doing so, we look forward to contributing to the development of the SNN community.
> > >
> > > - As for the completeness of the work, we deeply apologize for the oversight in experimental validation in our first manuscript. Fortunately, you point out our oversight during the review stage, and we further improve the manuscript based on this. We will include the additional experimental results and training details into the main text in future versions. In addition, although adding additional experiments during the rebuttal phase requires considerable effort, it is our responsibility to revise the work based on constructive suggestions! We are deeply grateful to you and the other reviewers for acknowledging the effort we have dedicated to this phase.
> > >
> > > *Suggestions for future work:*
> > >
> > > In our future work, **in terms of methodology**, we will explore more innovative approaches to make contributions to the advancement of the SNN field. **In terms of experimental design**, we will carefully study the experimental design of Fast-SNN [1], thereby fully verifying the effectiveness of future work from multiple aspects (diversity of tasks and architectures). **In terms of manuscript writing**, we will ensure that the experimental details and results, such as pruning rate, #SOP, and power consumption, are thoroughly presented in the main text.
> > >
> > > Yes, we will carefully improve this manuscript in future versions by incorporating additional experiments into the main text. Thank you for your professional and constructive feedback, which has been crucial in enhancing the quality of our paper! More importantly, your suggestions have been invaluable in guiding my future research! Once again, we would like to express my sincere gratitude for your thoughtful suggestions! We wish you continued success in all your future endeavors!
> > >
> > > Best regards.
> > >
> > > > *[1] Fast-SNN: fast spiking neural network by converting quantized ANN. IEEE T-PAMI.*

---

### Official Review · Reviewer_rh3C · 2024-11-03

**Soundness:** 3
**Presentation:** 3
**Contribution:** 3
**Rating:** 8
**Confidence:** 5

**Summary:**

This work focuses on the applicability of SNNs on edge devices, integrating weights quantization and structured pruning techniques, and providing an in-depth analysis of the main challenges in each respective area. Specifically, for weights quantization, the work proposes weight scaling; for structured pruning, it introduces a method to remove redundant convolutional kernels by analyzing the temporal-spatial spike activity singular values in the spike feature matrix. This is an effective joint exploration of weight quantization and pruning in SNNs. I appreciate the idea of leveraging quantization, pruning, and event-driven techniques—three approaches that accelerate, conserve energy, and enhance the efficiency of neural networks.

**Strengths:**

1. The illustrations are thoughtfully crafted.

2. The manuscript presents a clear structure, and maintains logical consistency.

3. A detailed analysis of the weight distribution in SNNs provides the motivation for the re-scaling method.

4. The experiments demonstrate the effectiveness of this method on both static and dynamic datasets.

**Weaknesses:**

1. Regarding the selection of the quantization scaling factor $\gamma$, this work employs three different sampling methods. Although experiments indicate that scaling with the 1-norm mean yields the best results, I believe there is still room for improvement, such as using a piecewise $\gamma$. I encourage the authors to provide further explanation.

2. To my knowledge, when two or more model lightweigh techniques are employed, compatibility issues often arise, such as the order of applying these techniques and the training strategies involved. Additional clarification is needed on these aspects.

3. At the end of Algorithm. 1, fine-tuning is required after pruning. The specific details of this fine-tuning process should be provided.

4. In Table. 2 of the experimental section, the baselines for A-F are not consistent. I suggest the authors provide a unified ablation study. Additionally, regarding the compatibility issues of interest, I encourage the authors to conduct an ablation study on the compatibility of quantization and pruning.

**Questions:**

1.To my knowledge, the highest-performing architecture in the SNN field is the Spiking Transformer [1-5]. Please discuss whether the proposed method can be effectively applied to the Spiking Transformer. If feasible, please provide some preliminary experimental results.

[1] Zhou, Z., Zhu, Y., He, C., Wang, Y., Shuicheng, Y. A. N., Tian, Y., & Yuan, L. Spikformer: When Spiking Neural Network Meets Transformer. In The Eleventh International Conference on Learning Representations.

[2] Yao, M., Hu, J., Zhou, Z., Yuan, L., Tian, Y., Xu, B., & Li, G. (2024). Spike-driven transformer. Advances in neural information processing systems, 36.

[3] Zhou, C., Yu, L., Zhou, Z., Ma, Z., Zhang, H., Zhou, H., & Tian, Y. (2023). Spikingformer: Spike-driven residual learning for transformer-based spiking neural network. arXiv preprint arXiv:2304.11954.

[4] Zhou, Z., Che, K., Fang, W., Tian, K., Zhu, Y., Yan, S., ... & Yuan, L. (2024). Spikformer v2: Join the high accuracy club on imagenet with an snn ticket. arXiv preprint arXiv:2401.02020.

[5] Yao, M., Hu, J., Hu, T., Xu, Y., Zhou, Z., Tian, Y., ... & Li, G. Spike-driven Transformer V2: Meta Spiking Neural Network Architecture Inspiring the Design of Next-generation Neuromorphic Chips. In The Twelfth International Conference on Learning Representations.

**Details Of Ethics Concerns:**

None.

---

> ### Author Response · Authors · 2024-11-23
> **Response to W2**
>
> > *W2: To my knowledge, when two or more model lightweigh techniques are employed, compatibility issues often arise, such as the order of applying these techniques and the training strategies involved. Additional clarification is needed on these aspects.*
>
> *Response to W2:* Thank you for pointing out this issue. We have adopted the "quantize first, then prune" strategy based on the following two considerations.
>
> - **First, this strategy can better guarantee the effect of pruning technique.** Specifically, if pruning is applied before quantization, important convolutional kernels identified in the full-precision parameter domain may become misaligned after quantization, as the quantization reintroduces additional errors. In contrast, by quantizing first and then pruning, redundant convolutional kernels are identified directly in the target low-precision parameter domain. This order allows for more accurate identification and preservation of critical kernels.
>
> - **Second, this strategy significantly reduces training overhead.** Pruning before quantization requires three weight updates: "full-precision SNN training→ pruning with fine-tuning → quantization with fine-tuning," while "quantize first, then prune" only requires two adjustments: "quatized SNN training→ pruning with fine-tuning."
>
> To more effectively address your concerns, we have **conducted additional validation experiments by reversing the order of quantization and pruning**. The experiments are conducted using the ResNet20 architecture on the CIFAR-100 dataset under the weight bit-width of 4, and the results are summarized in the table below. As shown in the following table, our ReScaW and SVS improve both performances irrespective of the order in which they are applied, yielding an 1.83% improvement in PQ-SNN and a 4.46% improvement in QP-SNN. QP-SNN achieves the highest performance of 74.73%.
>
>
> | Method   | PQ-SNN baseline | PQ-SNN                         | QP-SNN baseline | QP-SNN                         |
> | -------- | --------------- | ------------------------------ | --------------- | ------------------------------ |
> | Accuracy | 71.51%          | 73.34% (PQ-SNN baseline+1.83%) | 70.27%          | 74.73% (QP-SNN baseline+4.46%) |

---

> ### Author Response · Authors · 2024-11-23
> **Response to W1**
>
> > *W1: Regarding the selection of the quantization scaling factor γ, this work employs three different sampling methods. Although experiments indicate that scaling with the 1-norm mean yields the best results, I believe there is still room for improvement, such as using a piecewise γ. I encourage the authors to provide further explanation.*
>
> *Response to W1:* Yes, your suggestion is correct. By using piecewise scaling factors, we can more precisely control the quantization levels of weights, thereby better capturing important information. For example, consider weights that follow a bell-shaped distribution. We can assign small scaling factors to weights concentrated around zero so that they are mapped to a larger range after quantization, thus using more allocated bitwidth. In contrast, we assign large scaling factors to the weights at both tails, so they are mapped to a small range and use less allocated bitwidth. **However, this piecewise scaling factors method essentially results in non-uniform quantization**, because employing different scaling factors in different intervals leads to uneven intervals of overall quantization levels.
>
> This non-uniform quantization typically may achieve higher accuracy for a fixed bit-width, but **it is difficult to deploy efficiently on general-purpose computing hardware** like GPUs and CPUs. Therefore, we do not consider piecewise scaling factors in our method. In contrast, our proposed ReScaW strategy enhances bit-width utilization while maintaining the hardware-friendly characteristics of uniform quantization, offering a simple, effective, and practical solution.

---

> ### Author Response · Authors · 2024-11-23
> **Response to W3**
>
> > *W3: At the end of Algorithm. 1, fine-tuning is required after pruning. The specific details of this fine-tuning process should be provided.*
>
> *Response to W3:* After carefully reviewing the appendix, we find that the training details for the fine-tuning stage are missing, and some hyperparameters during training are incorrectly described. We apologize for these oversights and summarize the correct training and fine-tuning parameter setups as follows.
>
> |    Hyper-parameter    |           CIFAR-10/100           | TinyImageNet  |   ImageNet    |  DVS-CIFAR10  |
> | :-------------------: | :------------------------------: | :-----------: | :-----------: | :-----------: |
> |         Epoch         | 300 (Training)/150 (Fine tuning) |    300/150    |    320/200    |    300/150    |
> |       Optimizer       |             SGD/Adam             |   SGD/Adam    |    SGD/SGD    |   SGD/Adam    |
> | Initial learning rate |            0.1/0.001             |   0.1/0.001   |   0.1/0.05    |   0.1/0.001   |
> |  Learning rate decay  |          Cosine/Cosine           | Cosine/Cosine | Cosine/Cosine | Cosine/Cosine |

---

> ### Author Response · Authors · 2024-11-23
> **Response to W4**
>
> > *W4: In Table. 2 of the experimental section, the baselines for A-F are not consistent. I suggest the authors provide a unified ablation study. Additionally, regarding the compatibility issues of interest, I encourage the authors to conduct an ablation study on the compatibility of quantization and pruning.*
>
> *Response to W4:* The ablation experiments in Table 2 are divided into two groups with different baselines. Models in group A-B use **only quantization**, while those in group C-F **use both quantization and pruning**. The A-B comparison is designed to show that ReScaW improves the performance of quantized models, demonstrating its effectiveness **even when used alone**. The C-F comparison is designed to show the performance improvement of ReScaW and SVS for **quantized and pruned** SNNs.
>
> To address your compatibility concerns, we conduct additional ablation experiments by switching the order of quantization and pruning. Experiments are performed on the CIFAR-100 dataset using a ResNet20 architecture. The results are summarized in the table below, from which two conclusions can be obtained.
>
> - **First,** the proposed ReScaW and SVS can improve both performance, regardless of the order in which they are applied, leading to an 1.83% improvement in PQ-SNN and a 4.46% improvement in QP-SNN.
> - **Second,** QP-SNN achieves the highest performance (surpassing PQ-SNN by 1.39%), which proves that ‘quantize first, then prune’ is a more effective approach. (For detailed theoretical analysis, see Response to W2.)
> | Method   | PQ-SNN baseline | PQ-SNN                         | QP-SNN baseline | QP-SNN                         |
> | -------- | --------------- | ------------------------------ | --------------- | ------------------------------ |
> | Accuracy | 71.51%          | 73.34% (PQ-SNN baseline+1.83%) | 70.27%          | 74.73% (QP-SNN baseline+4.46%) |

---

> ### Author Response · Authors · 2024-11-23
> **Response to Q1**
>
> > *Q1: To my knowledge, the highest-performing architecture in the SNN field is the Spiking Transformer. Please discuss whether the proposed method can be effectively applied to the Spiking Transformer. If feasible, please provide some preliminary experimental results.*
>
> *Response to Q1:* Thank you for your valuable suggestion. To address your concerns, we present additional experiments using Spikingformer [1] as an example to demonstrate that our QP-SNN can be effectively applied to Spiking Transformer architectures.
>
> Experimental details: We select the Spikingformer-4-384 structure and validate it on the CIFAR-100 dataset. The training process is as follows: we first train a 4-bit quantized Transformer with ReScaW strategy, and then we apply the SVS pruning criterion to identify and remove redundant convolutional kernels from the quantized Transformer, finally the pruned Transformer is fine-tuned to obtain the compressed version.
>
> Experimental results: The results are summarized in the table below. As shown, the QP-SpikingTransformer **achieves a** **87.93% reduction in model size, a 55.48% decrease in SOPs, and a 55.64% reduction in power consumption, while maintaining the satisfactory performance of 76.94%**. These results fully validate the effectiveness of QP-SNN for Spiking Transformers.
>
> | Architecture        | Method         | Connection | Bit Width | Model size(MB) | SOPs(M) | Power(mJ) | Accuracy |
> | ------------------- | -------------- | ---------- | --------- | -------------- | ------- | --------- | -------- |
> | Spikingformer-4-384 | Full-precision | 100%       | 16        | 18.64          | 292.14  | 0.266     | 79.09%   |
> | Spikingformer-4-384 | QP-SNN         | 44.74%     | 4         | 2.25           | 130.05  | 0.118     | 76.94%   |
> >*[1] Zhou, Chenlin, et al. Spikingformer: Spike-driven residual learning for transformer-based spiking neural network. arXiv preprint arXiv:2304.11954 (2023).*

---

> ### Comment · Reviewer_rh3C · 2024-11-24
>
> I sincerely appreciate the authors' detailed response, which has addressed my concerns. I am willing to raise my score.

---

> > ### Author Response · Authors · 2024-11-24
> >
> > We sincerely appreciate your recognition of our work. We will thoroughly revise the manuscript in accordance with your suggestions and those of the other reviewers to enhance its quality.

---

### Public Comment · ~Changze_Lv1 · 2025-04-19

Dear Authors,

Greetings! I greatly appreciate your work and believe it holds significant value in the field of quantization for Spiking Neural Networks. I also noticed your previous paper, Q-SNNs: Quantized Spiking Neural Networks, accepted at ACM MM 2024, which I find very forward-looking and insightful.

Unfortunately, I couldn't find the released code for either of these two papers. May I ask if you have any plans to make the code publicly available?

Thank you very much!

---

> ### Public Comment · ~Wenjie_Wei1 · 2025-04-19
>
> Dear Changze,
>
> Thank you for your interest and recognition of our work. I have organized the code and sent it to your email. I hope it will be helpful for your research.
>
> Best regards,
> Wenjie.

---

### Public Comment · ~Do_Thanh_Dat1 · 2025-09-03

Dear Authors,

Thank you very much for your excellent work on quantization and pruning for Spiking Neural Networks. I find your research very inspiring and would like to study it in more detail.
May I ask if there is a publicly available version of the source code, or if you plan to release it in the future?

Thank you for your time and support!

---

> ### Public Comment · ~Jieyuan_Zhang2 · 2025-09-03
>
> Dear Do Thanh Dat,
>
> Thank you for your interest in our work and your kind recognition. We are currently in the process of optimizing and organizing our research code for public release. We expect to make the latest version available within the next month.
>
> Please feel free to reach out if you have any questions about our work in the meantime.
>
> Best regards,
>
> Eric

---

### Public Comment · ~Linliang_Chen2 · 2025-11-24

Dear Authors,

I have read your paper on quantization and pruning for Spiking Neural Networks with great interest. The approach you presented is highly valuable to my current research.

I would be most grateful if you could share the implementation code with me at your convenience. If the code is not yet ready for distribution, I would appreciate any information about potential future release plans.

Thank you very much for your time.

Warm regards,

---

### Meta-Review · Area_Chair_PLh1 · 2024-12-19

**Metareview:**

This paper proposes a hardware-friendly and lightweight SNN, aimed at effectively deploying high-performance SNN in resource-limited scenarios. After the response, it receives one borderline reject, one borderline accept, and two accept. For reviewer Likb, though the rating is 5, the concerns are well addressed. The strengths of the paper, including the clear motivation, interesting ideas, and good results are well recognized. I agree with them and think the current manuscript meets the requirements of this top conference. Please also incorporate the response in the revised paper.

**Additional Comments On Reviewer Discussion:**

The response well addressed the concerns.

---

### Decision · Program_Chairs · 2025-01-22

Accept (Poster)